# Generative Visual Code Mobile World Models

**Woosung Koh** [* 1 2]  **Sungjun Han** [* 1]  **Segyu Lee** [1 2]  **Se-young Yun** [2]  **Jamin Shin** [1]

## Abstract

Mobile Graphical User Interface (GUI) World Models (WMs) offer a promising path for improving mobile GUI agent performance at train- and inference-time. However, current approaches face a critical trade-off: text-based WMs sacrifice visual fidelity, while the inability of visual WMs in precise text rendering led to their reliance on slow, complex pipelines dependent on numerous external models. We propose a novel paradigm: *visual world modeling via renderable code* generation, where a *single* Vision-Language Model (VLM) predicts the next GUI state as executable web code that renders to pixels, rather than generating pixels directly. This combines the strengths of both approaches: VLMs retain their linguistic priors for precise text rendering while their pre-training on structured web code enables high-fidelity visual generation. We introduce gWorld (8B, 32B), the first open-weight visual mobile GUI WMs built on this paradigm, along with a data generation framework that automatically synthesizes code-based training data. In extensive evaluation across **4** in-distribution and **2** out-of-distribution benchmarks, gWorld sets a new *pareto frontier* in accuracy versus model size, outperforming **8** frontier open-weight models over **50.25×** larger. Further analyses show that (1) scaling training data yields meaningful gains, (2) each component of our pipeline improves data quality, and (3) stronger world modeling improves downstream mobile GUI policy performance.

🌐 **Project Page**    ⭘ **Code**
🤗 **gWorld (8B, 32B)**    🤗 **MWMBench**

## 1. Introduction

Improving policy performance on mobile Graphical User Interface (GUI) tasks has become a rapidly expanding research

---
[*]Equal contribution  [1]Trillion Labs [2]KAIST AI. Correspondence to: Se-young Yun <yunseyoung@kaist.ac.kr>, Jamin Shin <jay@trillionlabs.co>.

*Proceedings of the 43$^{rd}$ International Conference on Machine Learning*, Seoul, South Korea. PMLR 306, 2026. Copyright 2026 by the author(s).

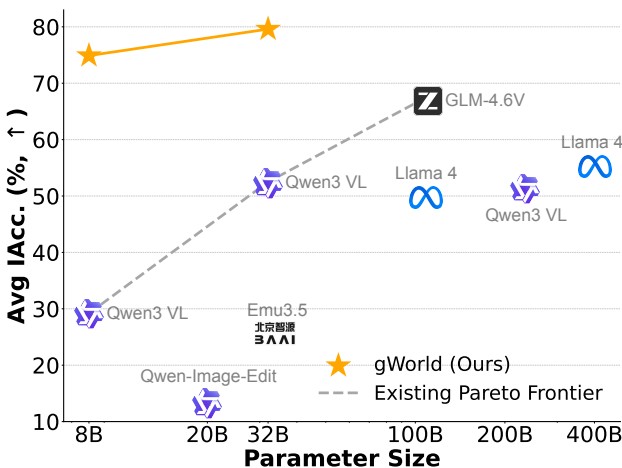

*Figure 1.* **Average Instruction Accuracy (IAcc.) across all six benchmarks.** gWorld 8B and 32B achieve a new pareto frontier in terms of model size (log$_{10}$ scaled). The existing pareto frontier was defined by Qwen3 VL 8B, 32B, and GLM 4.6V 106B. Notably, extremely large models (e.g., Llama 4 402B) do not reach this pareto frontier, while text-image-to-image models (e.g., Emu3.5 34B) struggle with mobile GUI dynamics.

area (Wang et al., 2024; Ye et al., 2025; Zhang et al., 2025; Li et al., 2025a; Nguyen et al., 2025; Liu et al., 2025; Niu et al., 2025), driven by the ubiquitous nature of mobile computing, with an estimated 8.9 billion mobile subscriptions worldwide (Ericsson, 2025). An emerging line of literature demonstrates that leveraging a generative mobile World Model (WM) to predict future states can significantly enhance policy performance during both training (Fang et al., 2025; Wang et al., 2025) and inference (Chae et al., 2025; Gao et al., 2025; Gu et al., 2025; Li et al., 2025b; Cao et al., 2026).

While these approaches yield substantial gains, they predict the next state in text; an abstraction over the pixel-space GUI state. This abstraction discards critical GUI information, including fine-grained spatial layout and visual attributes (e.g., iconography, typography, and color) (Chae et al., 2025; Luo et al., 2025; Cao et al., 2026). Moreover, text-only world representations limit Vision-Language Model (VLM)-based policies, which have been shown to outperform language-only models on mobile GUI tasks (Hong et al., 2024; Lu et al., 2024; Gou et al., 2025). In response, VIMO (Luo et al., 2025) introduced the first visual mobile GUI WM and showed that visual world modeling yields larger policy

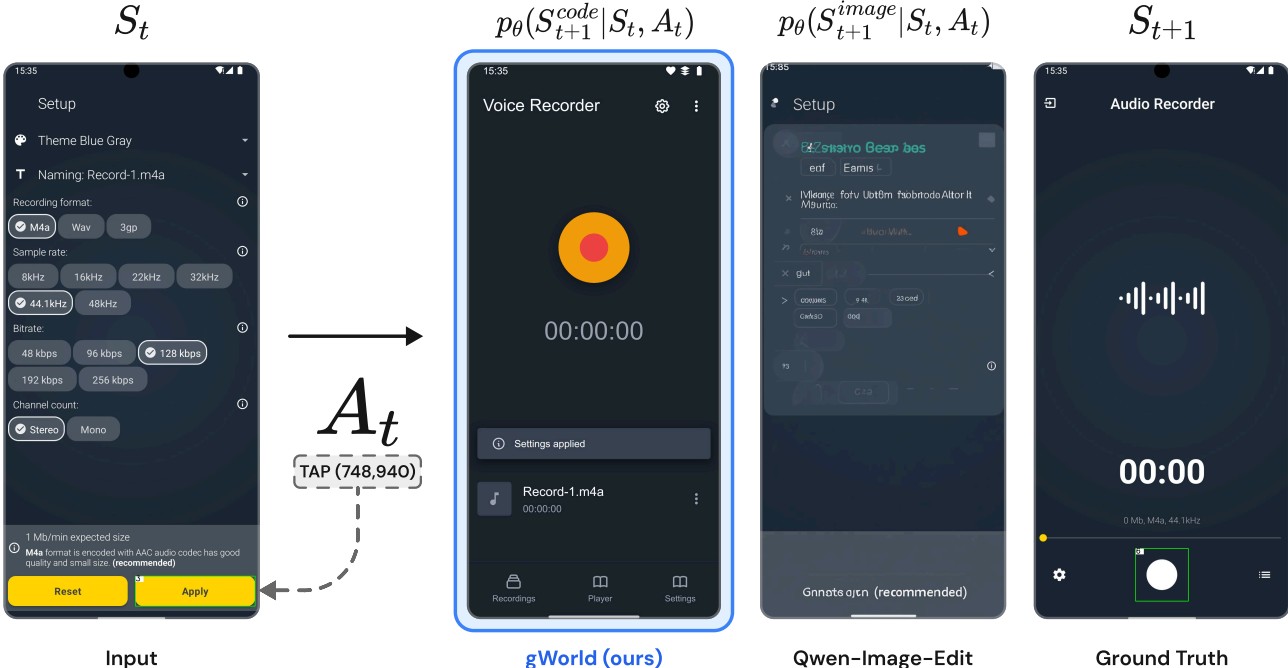

$$S_t \qquad p_\theta(S_{t+1}^{code}|S_t, A_t) \qquad p_\theta(S_{t+1}^{image}|S_t, A_t) \qquad S_{t+1}$$

$A_t$

TAP (748,940)

Input · gWorld (ours) · Qwen-Image-Edit · Ground Truth

*Figure 2.* **Mobile GUI world modeling via renderable code.** Given an image state $S_t$ and action $A_t$, the model predicts the next state $S_{t+1}$. Our model, gWorld, generates renderable web code to ensure pixel-perfect text and structurally accurate layouts. In contrast, image-gen baselines (e.g., Qwen-Image-Edit 20B) struggle with the discrete nature of GUIs, frequently producing illegible text and distorted layouts. See Appendix Fig. 13, 14, 15 for additional qualitative examples.

improvements than text-based alternatives.

However, we observe three notable disadvantages of VIMO. First, VIMO relies on a complex multi-stage pipeline rather than a single self-contained model, resulting in significant computational overhead and latency (Luo et al., 2025; Cao et al., 2026). Concretely, their framework uses (1) an external Optical Character Recognition (OCR) model for text detection, (2) box-based text masking, (3) an external frontier VLM (GPT-4o) to filter masked regions, (4) a custom-trained diffusion model to generate the next-state image, and (5) two additional GPT-4o calls to fill in next-state text. Second, their formulation converts coordinate-based actions into natural-language instructions via GPT-4o, effectively outsourcing visual grounding to a closed-weight model. Lastly, VIMO does not release the weights of its custom-trained diffusion model, making the system difficult to reproduce and deploy.

**Contribution.** In response, we present gWorld (8B, 32B), which to our knowledge, are the first open-weight, single self-contained world models specialized for visual mobile GUI world modeling that operates via *renderable code generation*. We start by analyzing the limitations of using a image-generation model for mobile GUI World Models (§ 2.2), with further detailed analysis in § 4.3. To alleviate this, we show for the *first* time that a *code-based*

*representation* can be leveraged for mobile GUI World Models (§ 2.3). As there are no code-based GUI world modeling training datasets, we present our data generation framework (§ 2.4). Specifically, we repurpose offline mobile-agent trajectories into $(S_t, A_t)$-conditioned next-state pairs, automatically converts $S_{t+1}$ from pixels to renderable web code, and adds free look-ahead reasoning traces, producing large-scale SFT data for training code-generating GUI WMs. Furthermore, due to the lack of comprehensive visual mobile GUI world modeling benchmarks, we present MWMBENCH, curating and open-sourcing *four* in- and *two* out-of-distribution (OOD) benchmarks to evaluate mobile GUI WMs (§ 3, Tab. 1). We empirically demonstrate that our data generation framework and model is effective:

- Our models outperform **8** frontier open-weight image- and code-generation models up to **50.25×** larger, across **6** in- and out-of-distribution benchmarks (§ 4.2, Fig. 1, Tab. 2).

- We demonstrate that scaling our dataset size (37K, 77K, 129K, 240K) leads to predictable gains, closely following a power law (§ 4.4, Fig. 5).

- Ablation studies demonstrate that each component of our method contributes meaningfully (§ 4.5, Tab. 3, Fig. 6).

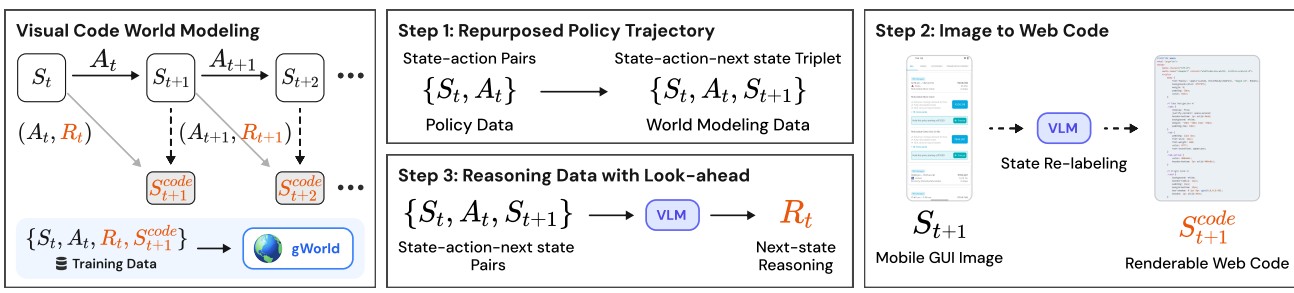

*Figure 3.* **Schematic diagram of our data generation pipeline.** We construct VLM world modeling data via three steps: **(1)** Repurposing offline policy trajectories into transition triplets; **(2)** Cross-modal relabeling of the ground-truth next state from pixels ($S_{t+1}^{\text{image}}$) to renderable web code ($S_{t+1}^{code}$); and **(3)** Synthesizing reasoning traces ($R_t$) using look-ahead access to the target state. The final training objective is to predict both the reasoning trace and the code-based next state: $(S_t, A_t) \to (R_t, S_{t+1}^{\text{code}})$. For visual succinctness we denote $S_t^{\text{image}}$ as $S_t$ without the superscript in the diagram.

- Finally, we experimentally demonstrate that improved WM performance translates to policy model performance gains (§ **4.8**, Tab. **6**).

## 1.1. Related Work

**Code-based World Models.** Dainese et al. (2024); Copet et al. (2025) study code-based WMs primarily to improve code-generation tasks. Alternatively, code-based WMs have been applied to different domains like game playing and fictional worlds (Feng et al., 2025; Lehrach et al., 2025). However, none study code-based world models for mobile GUI world modeling.

**Image to Web Code.** Prior work investigates converting web page images into web code to automate front-end development (Yun et al., 2024; Gui et al., 2025; Wan et al., 2025; Jiang et al., 2025; Si et al., 2025; Leviathan et al., 2025). In contrast, we show that modern VLMs can be trained to reconstruct complex mobile GUIs (e.g., settings screens and camera UIs) via web code generation alone.

**Mobile UI Simulator.** We discuss UISim (Xiang et al., 2025), which employs a multi-stage mobile GUI generation pipeline similar to VIMO, combining a VLM with a diffusion-based image model. However, the authors provide minimal implementation details necessary for replication, and the model weights remain closed source. Critically, their evaluation is limited to visual fidelity within a single in-distribution setting, lacking assessment of world modeling dynamics; i.e., action-contingent transitions.

## 2. `gWorld`: Generative Visual Code World Modeling

### 2.1. Problem Setting

In a Markov Decision Process (MDP; (Bellman, 1957)), a WM corresponds to the transition distribution $p : \mathcal{S} \times \mathcal{A} \to$ $\Delta(\mathcal{S})$, where $\mathcal{S}$ and $\mathcal{A}$ denote the state and action spaces and $\Delta(\mathcal{S})$ is the probability simplex over next states. Our goal is to train a generative WM parameterized by $\theta$, i.e., $p_\theta(S_{t+1} \mid S_t, A_t)$ (Fig. 2). For supervised fine-tuning (SFT; Wei et al. (2022a)), we denote the input as $X$ and the target label as $Y$.

### 2.2. Motivation: Limitation of Generating Pixel-based Next State.

As shown in Fig. 2 and 13, while frontier open-weight models reconstruct layouts resembling the input ($S_t$), they often fail to generate plausible next states ($S_{t+1}$) respecting transition dynamics. Specifically, they frequently produce illegible text (Luo et al., 2025) and struggle with states requiring novel layouts. We empirically corroborate these qualitative observations in § **4.3**.

### 2.3. Next World States as Renderable Code

To overcome the limitations of direct pixel generation, we emulate mobile GUI world modeling with structured web code. More specifically, we post-train VLMs to generate next states in web code. VLMs are well suited for modeling GUI transitions due to their linguistic priors and broad world knowledge. First, VLMs can generate precise, legible text, which remains a major bottleneck for image-gen models (see Fig. 2, 13). Futhermore, they can synthesize semantically coherent linguistic content aligned with the application context. For example, when predicting the next state of an email app, a GUI world model should render an interface populated with contextually plausible, *realistic* email content (see Fig. 2, 13, 14, 15). Finally, the prevalence of structured web code in VLM pre-training provides a strong inductive bias, making VLMs a natural foundation for generative visual code GUI world models.

| Benchmark | World Modality | Action Space | Dataset Composition | |
|---|---|---|---|---|
| | | | In-Distribution | OOD |
| MobileWorldBench | Text | Converted to Text | 2× (AitW, AC) | ✗ |
| VIMO's Benchmark | Visual | Converted to Text | 2× (AitW, AC) | ✗ |
| MWMBENCH (ours) | Visual | Original Coordinates and Text | 4× (AitW, GUIO, AC, AMEX) | 2× (AW, KA) |

*Table 1.* **Comparison with existing mobile world modeling benchmarks.** Prior benchmarks simplify the problem by converting actions to text and testing only in-distribution. In contrast, MWMBENCH allows evaluations on the **native visual action space** (preserving original coordinates) and is the first to assess **zero-shot generalization** on held-out out-of-distribution sets. *Datasets:* Android in the Wild (AitW), GUIOdyssey (GUIO), AndroidControl (AC), Android Multi-annotation Expo (AMEX), AndroidWorld (AW), and KApps (KA).

## 2.4. World Model Training Data Generation

Our framework generates VLM SFT (Liu et al., 2023) data of the form $\{(X : (S_t^{\text{image}}, A_t), Y : (R_t, S_{t+1}^{\text{code}}))\}_{t=1}^{T-1}$. Here $R_t$ is a text-based reasoning trace (Wei et al., 2022b), included because reasoning supervision is a well-established way to improve VLM performance (Rose et al., 2023; Chen et al., 2024c; Zhang et al., 2024; Chen et al., 2024b). **(1)** First, we repurpose abundant offline policy trajectory data as WM training data. **(2)** Second, we convert the next-state supervision ($S_{t+1}$) from pixels to renderable web code. **(3)** Finally, we synthesize reasoning traces ($R_t$) using free look-ahead to the ground-truth next state. We provide a visual diagram in Fig. 3, and pseudocode in Appendix A.

**(1) Repurposing Policy Trajectory as World Modeling Data.** We first repurpose existing large-scale mobile agent *policy* trajectory data into world modeling data. Given an episode trajectory $\{(X : S_t^{\text{image}}, Y : A_t)\}_{t=1}^{T}$, we synthesize transition examples $\{(X : S_t^{\text{image}}, A_t, Y : S_{t+1}^{\text{image}})\}_{t=1}^{T-1}$, matching the WM objective $p_\theta(S_{t+1}|S_t, A_t)$. This transformation reduces the number of examples per episode from $T$ to $T - 1$.

**(2) Synthetic Cross-modal State Re-labeling.** As our VLM outputs text, we re-label the next-state target $Y$ from pixels to renderable web code. Leveraging a frontier model $\pi^*$ with strong image-to-web-code capabilities, we obtain $S_t^{\text{code}} \leftarrow \pi^*(S_t^{\text{image}}, P^{\text{img-to-code}})$; the prompt $P^{\text{img-to-code}}$ is provided in Appendix A.

**(3) Reasoning Data with Free Look-ahead.** As we have access to the ground truth next state ($S_{t+1}$) for $S_t$, we generate reasoning traces $R_t$ with look-ahead access to $S_{t+1}$. The look-ahead grounds $R_t$ in the true next-state transition, ensuring alignment between the reasoning trace and $S_{t+1}^{code}$. For the WM, the introduction of $R_t$ decomposes the complex world modeling problem into two simpler sub-problems: first predicting state changes in natural language, then converting this description to web code. Concretely, we use frontier model $\pi^*$ to synthesize $R_t \leftarrow \pi^*(S_t^{\text{image}}, A_t, S_{t+1}^{\text{image}}, P^{\text{look-ahead}})$; the prompt

$P^{\text{look-ahead}}$ is provided in Appendix A.

## 2.5. World Model Training.

We generate a dataset of 260K samples using our method, derived from existing policy trajectories in Android in the Wild (AitW; Rawles et al. (2023)), GUIOddyssey (GUIO; Lu et al. (2025)), AndroidControl (AC; Li et al. (2024)), and Android Multi-annotation Expo (AMEX; Chai et al. (2025)). The frontier model ($\pi^*$) used for data generation experiments is `Gemini 3 Flash`. The base models for training are `Qwen3 VL 8B` and `32B` (Bai et al., 2025), selected as they represent the frontier of open-weight VLMs. We validate our post-training data on frontier open-weight models as we can examine whether our proposed dataset is *novel* and *useful*, given a good base model. Hyperparameters and further details are available in Appendix C.

## 3. MWMBENCH: Comprehensive Mobile GUI World Modeling Benchmark

We introduce Mobile World Model Bench (MWMBENCH), a comprehensive benchmark for evaluating world modeling in mobile GUI environments. MWMBENCH, consisting of $(S_t, A_t, S_{t+1})$ tuples from 6 data sources, enables systematic measurement of next-state prediction quality $\hat{S}_{t+1}$. It represents real-world mobile GUI usage, spanning diverse applications, tasks, and interaction patterns in different languages. Furthermore, MWMBENCH addresses *three* critical limitations of existing mobile GUI WM benchmarks that ensures close alignment with real-world deployment scenarios (see Tab. 1). Details are in Appendix B.

**Visual World Modeling.** First, unlike MobileWorldBench (Li et al., 2025b) which can only evaluate text-based WMs, we evaluate world models in the native visual modality so that rich GUI details and semantics are preserved.

**Real-world Action Space.** Second, unlike MobileWorldBench and VIMO's benchmark, we keep actions in coordinate space rather than converting them to text, avoiding dependence on an external frontier model (Luo et al., 2025).

| | Image-gen | | Code-gen | | | | | | | |
|---|---|---|---|---|---|---|---|---|---|---|
| **Model:** | `Qwen-I-E` | `Emu3.5` | `Llama 4` | | `8B` | `Qwen3 VL` | | `GLM-4.6V` | `gWorld` | |
| **Parameter Size:** | `20B` | `34B` | `109B-A17B` | `402B-A17B` | | `32B` | `235B-A22B` | `106B` | `8B` | `32B` |
| **MWMBENCH-AITW** | | | | | | | | | | |
| IAcc. (%, ↑) | 15.4 | 23.4 | 47.6 | 47.2 | 21.5 | 46.8 | 36.1 | 60.9 | 68.8 | **71.7** |
| └ Render Fail (%, ↓) | — | — | 4.4 | 9.4 | 33.8 | 11.6 | 40.0 | 2.4 | 0.8 | **0.6** |
| Similarity (%, ↑) | 60.1 | **68.7** | 57.9 | 58.9 | 49.9 | 59.0 | 62.9 | 64.7 | 66.3 | 67.3 |
| **MWMBENCH-GUIODYSSEY** | | | | | | | | | | |
| IAcc. (%, ↑) | 13.0 | 25.8 | 53.1 | 55.8 | 28.2 | 52.0 | 54.7 | 68.2 | 77.2 | **81.5** |
| └ Render Fail (%, ↓) | — | — | 1.2 | 7.8 | 51.4 | 16.0 | 27.2 | 3.8 | 1.2 | **0.8** |
| Similarity (%, ↑) | 63.8 | 68.8 | 62.3 | 64.0 | 48.3 | 62.7 | 69.7 | 72.5 | 73.3 | **73.7** |
| **MWMBENCH-ANDROIDCONTROL** | | | | | | | | | | |
| IAcc. (%, ↑) | 11.7 | 27.7 | 50.7 | 58.6 | 31.1 | 53.2 | 51.9 | 74.2 | 78.4 | **82.9** |
| └ Render Fail (%, ↓) | — | — | 1.0 | 8.6 | 42.8 | 13.4 | 34.2 | 1.4 | 2.6 | **0.8** |
| Similarity (%, ↑) | 63.8 | 68.6 | 61.4 | 63.1 | 53.4 | 64.1 | 68.8 | 71.4 | 72.8 | **74.2** |
| **MWMBENCH-AMEX** | | | | | | | | | | |
| IAcc. (%, ↑) | 10.9 | 21.7 | 49.0 | 58.3 | 33.7 | 56.9 | 51.2 | 69.5 | 82.6 | **86.1** |
| └ Render Fail (%, ↓) | — | — | 0.6 | 12.6 | 31.6 | 3.8 | 30.0 | 1.2 | 0.8 | **0.4** |
| Similarity (%, ↑) | 64.4 | 71.6 | 66.9 | 68.1 | 59.2 | 70.0 | 71.7 | 73.2 | 74.3 | **75.4** |
| **MWMBENCH-ANDROIDWORLD** (out-of-distribution) | | | | | | | | | | |
| IAcc. (%, ↑) | 13.8 | 29.1 | 51.0 | 54.3 | 30.8 | 53.4 | 51.1 | 74.1 | 75.0 | **79.9** |
| └ Render Fail (%, ↓) | — | — | 2.9 | 14.4 | 42.3 | 13.1 | 30.0 | 1.9 | 2.3 | **0.4** |
| Similarity (%, ↑) | 67.9 | **74.2** | 61.7 | 61.8 | 49.9 | 61.2 | 65.2 | 72.2 | 69.2 | 71.6 |
| **MWMBENCH-KAPPS** (out-of-distribution) | | | | | | | | | | |
| IAcc. (%, ↑) | 15.7 | 26.8 | 48.4 | 59.9 | 30.1 | 52.5 | 64.2 | 57.4 | 67.4 | **75.7** |
| └ Render Fail (%, ↓) | — | — | 1.8 | 2.2 | 38.8 | 8.1 | 15.4 | 4.4 | 0.8 | **0.6** |
| Similarity (%, ↑) | 71.0 | **71.2** | 57.3 | 58.6 | 50.5 | 62.8 | 67.3 | 63.7 | 66.1 | 66.2 |
| **Average** | | | | | | | | | | |
| IAcc. (%, ↑) | 13.4 | 25.8 | 50.0 | 55.7 | 29.2 | 52.5 | 51.5 | 67.4 | 74.9 | **79.6** |
| └ Render Fail (%, ↓) | — | — | 2.0 | 9.2 | 40.1 | 11.0 | 29.5 | 2.5 | 1.4 | **0.6** |
| Similarity (%, ↑) | 65.2 | 70.5 | 61.2 | 62.4 | 51.8 | 63.3 | 67.6 | 69.6 | 70.3 | **71.4** |

*Table 2*. **Main mobile world modeling results.** We compare `gWorld` against frontier image-generation and VLM baselines across in-distribution and OOD benchmarks. The best scores are **bolded** and the second best are underlined. `gWorld` 8B, 32B establishes a new pareto frontier, consistently outperforming significantly larger models (e.g., `Llama 4 402B-A17B`, `Qwen3-VL 235B-A22B`). Notably, our code-based approach virtually eliminates structural errors (<1% Render Fail), driving a **+45.7%** and **+27.1%** gain in average Instruction Accuracy (IAcc.) over the base models `Qwen3 VL 8B, 32B`, respectively.

This makes the evaluated world models directly compatible with real-world mobile execution, where actions are issued in coordinate space (Rawles et al., 2025).

**In- and Out-of-distribution Evaluations.** Lastly, we support four in-distribution (ID) and two out-of-distribution (OOD) evaluation for comprehensive assessment. For ID evaluation, we randomly sample 500 world modeling instances (§ 2.4 (1)) from trajectories in Android in the Wild (AitW; Rawles et al. (2023)), GUIOddyssey (GUIO; Lu et al. (2025)), AndroidControl (AC; Li et al. (2024)), and Android Multi-annotation Expo (AMEX; Chai et al. (2025)) to form held-out test sets. We designate these as ID because they are the training sets of prior works and ours.

To evaluate OOD generalization, we curate two new benchmarks: ANDROIDWORLD (AW) and KAPPS (KA). For ANDROIDWORLD, we automatically collect offline trajectories from AndroidWorld (Rawles et al., 2025) and convert them into world modeling tasks. For KAPPS, we manually curate extensive ground truth policy trajectories in Korean, reflecting Korea-centric mobile usage, and convert them into world modeling tasks. We designate these as OOD as no corresponding training datasets are publicly available. Details of these assets are provided in Appendix B, Tab. 8.

## 4. Empirical Study

### 4.1. World Modeling Experiment Set-up

**Evaluation Metric.** Following Luo et al. (2025), we use (**1**) Instruction Accuracy (IAcc.) and (**2**) a similarity score

against ground truth. We conduct a small-scale human study to verify that our metrics reflect human preferences in § 4.6.

**(1) IAcc.** This is our primary metric: a VLM-as-a-Judge that outputs a binary pass/fail verdict on whether the generated next state is consistent with the current state–action pair. IAcc. ← $\pi^*(S_t, A_t, \hat{S}_{t+1}; P^{\text{IAcc.}})$, where $\hat{S}_{t+1}$ is generated by our model and $P^{\text{IAcc.}}$ is the evaluation prompt (Appendix C). IAcc. measures action-conditioned next-state correctness, directly reflecting world-modeling performance. IAcc. has been extensively studied to correlate highly with humans in Luo et al. (2025). To mitigate judge-model (family) bias (Chen et al., 2024a; Panickssery et al., 2024; Li et al., 2026), we compute IAcc. as the mean of the binary verdicts from three frontier VLM judges: `GPT-5 Mini`, `Claude 4.5 Haiku`, and `Gemini 3 Flash`. Per-judge IAcc. scores are reported in Appendix Tab. 12, and we observe high inter-judge agreement (see Appendix Fig. 11, 12). To reduce computational overhead, we use a rule-based filter that identifies un-renderable web code which classifies these faulty instances as automatic failures prior to evaluation.

**(2) Similarity**. Embedding similarity reports the cosine similarity between the respective embeddings of the generated next-state image $\hat{S}_{t+1}$ and the ground-truth next-state image $S_{t+1}$. This metric captures perceptual similarity but does not verify action-conditioned semantic correctness (i.e., whether the transition matches the instruction). We report the average value of DINO v1 (Caron et al., 2021) and v2 (Oquab et al., 2024) vision encoders' embeddings. The granular results for each encoder is organized in Appendix Tab. 12.

**Baseline Models.** To the best of our knowledge, this work is the first to propose a unified visual world model for mobile GUIs; consequently, there are no specialized baselines directly comparable to our approach. We therefore benchmark against widely adopted frontier open-weight models (Maslej et al., 2025). We select frontier open-weight models released in the past year, including text-image-to-image models `Qwen-Image-Edit 20B` (Wu et al., 2025) and `Emu3.5 34B` (Cui et al., 2025), as well as VLMs including `Llama 4` (109B, 402B) (Meta, 2025), `Qwen3 VL` (8B, 32B, 235B) (Bai et al., 2025), and `GLM-4.6V 106B` (Team et al., 2025).

### 4.2. World Modeling Results

**Strong In/Out-of-Distribution Results against Existing Models.** `gWorld 32B` and `8B` achieve the best and second-best performance (IAcc.), respectively, across all six benchmarks (see Tab. 2). Notably, `gWorld` demonstrates robust generalization; its performance on OOD benchmarks does not significantly degrade compared to in-distribution settings. The next best-performing baselines

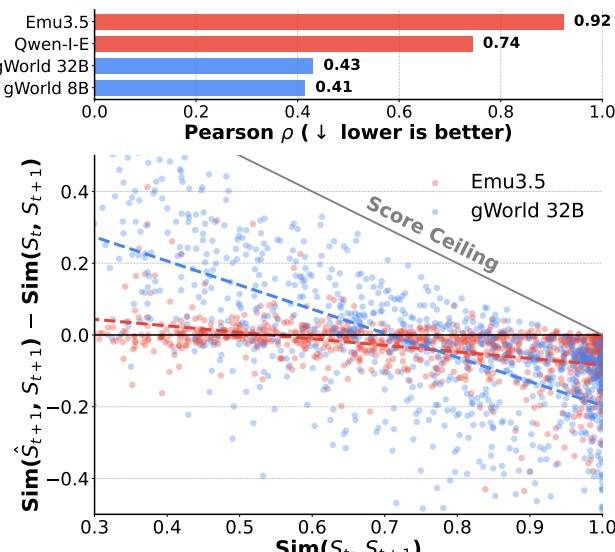

*Figure 4.* Correlation between input-output similarity and model performance. **Top:** Pearson correlation $\rho$ between $\text{Sim}(S_t, S_{t+1})$ and $\text{Sim}(\hat{S}_{t+1}, S_{t+1})$. Image generation models show strong positive correlations ($\rho > 0.7$), suggesting output quality largely depends on how similar $S_t$ and $S_{t+1}$ already are. **Bottom:** $\text{Sim}(\hat{S}_{t+1}, S_{t+1}) - \text{Sim}(S_t, S_{t+1})$ vs. $\text{Sim}(S_t, S_{t+1})$, with the gray line indicating the score ceiling. `Emu3.5 34B` clusters near zero, implying $\text{Sim}(\hat{S}_{t+1}, S_{t+1}) \approx \text{Sim}(S_t, S_{t+1})$; i.e., outputs nearly identical to inputs, $S_t \approx \hat{S}_{t+1}$. In contrast, `gWorld 32B` shows a wide vertical spread, indicating active state transformation with many samples achieving large positive gains toward the ceiling. Same analysis with `Qwen-Image-Edit 20B` is available in Appendix Fig. 21 with equivalent results.

are `GLM-4.6V 106B` and `Llama 4 402B-A17B`. This is particularly notable given that these models exceed the parameter count of `gWorld 8B` by factors of 13.25× and 50.25×, respectively. Moreover, `gWorld 32B` and `8B` rank first in terms of Similarity, with the exception of two benchmarks with `Emu3.5 34B` being the highest.

### 4.3. Further Analysis: Limitation of Image-gen Models

While image-generation baselines attain high visual similarity scores, they fail to capture mobile dynamics, achieving only 10.9 to 29.1% IAcc. (Tab. 2). We attribute this discrepancy to the high visual redundancy in mobile GUIs, where transitions often involve minimal changes (e.g., typing). Consequently, image-generation models can maximize similarity metrics by learning a trivial identity mapping—copying $S_t$ with minor edits—rather than modeling the semantic state transition to $S_{t+1}$.

Fig. 4 confirms that image-generation models' performance relies on input-output similarity rather than action understanding. Image-generation models exhibit strong Pearson correlations between the ground-truth transition similarity $\text{Sim}(S_t, S_{t+1})$ and their output similarity $\text{Sim}(\hat{S}_{t+1}, S_{t+1})$

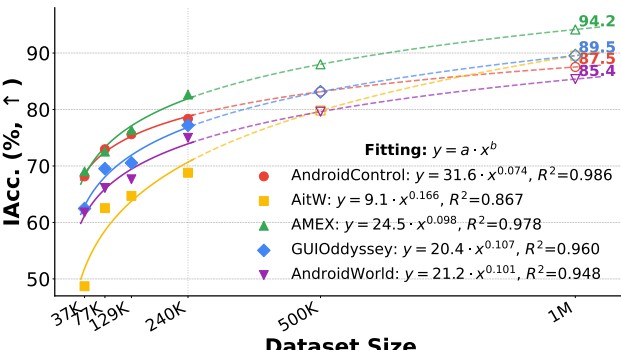

*Figure 5.* **Data scaling laws for mobile world modeling at 8B.** We fit power-law curves ($y = ax^b$) to the test performance across five distinct benchmarks as a function of training dataset size. The high coefficients of determination ($R^2 \geq 0.94$ for most splits) indicate a **predictable** and **non-saturating** relationship between data scale and performance. This suggests that our data generation pipeline has not yet reached its upper bound and will continue to improve with larger-scale repurposed trajectories.

(`Emu3.5 34B`: $\rho = 0.92$, `Qwen-Image-Edit 20B`: $\rho = 0.74$), whereas `gWorld` shows a much weaker correlation ($\rho \approx 0.4$). Furthermore, plotting the similarity gain (Fig. 4, bottom) reveals that `Emu3.5 34B` consistently yields near-zero values regardless of difficulty, implying the output $\hat{S}_{t+1}$ remains nearly identical to $S_t$. In contrast, `gWorld 32B` displays substantial variance, indicating it actively predicts structural changes based on the action rather than defaulting to a copying strategy.

### 4.4. Further Analysis: Scaling Data

To assess the scalability of our approach, we examine whether increasing the dataset size yields consistent performance improvements. We test data set sizes of 37K, 77K, 129K, and 240K and plot our training curves in Fig. 5; with further granular plots in Appendix Fig. 9 and 10. We observe monotonic performance gains as the dataset size increases, providing strong evidence of data quality and effective scaling.

Consistent with empirical scaling laws demonstrated in Li et al. (2024), Kaplan et al. (2020), Hoffmann et al. (2022), we observe that our dataset scaling follows a power law with an average $R^2$ of 0.948 (see Fig. 5). This analysis enables performance projection as we continue to generate data via our method. We compute the maximum number of trainable transitions attainable in Appendix C.1 based on the four existing offline trajectory data sets we use (AitW, GUIO, AC, and AMEX), and arrive at a maximum data set size of 3.7 million. Based on the power law trajectory in Fig. 5, we project significant performance gains by utilizing the remaining available trajectories.

| Metric | Alternative | Ours | Δ |
|---|---|---|---|
| Renderable Code (%, ↑) | 97 | 100 | **+3** |
| IAcc. (%, ↑) - `Gemini 3 Pro` | 94.60 | 100 | **+5.40** |
| IAcc. (%, ↑) - `Claude 4.5 Opus` | 84.80 | 86.70 | **+1.90** |

*Table 3.* **Ablation on $S_{t+1}^{\text{code}}$ train data quality.** Our method outperforms the naïve alternative, achieving perfect code renderability and higher IAcc. across strong VLM-as-a-Judges.

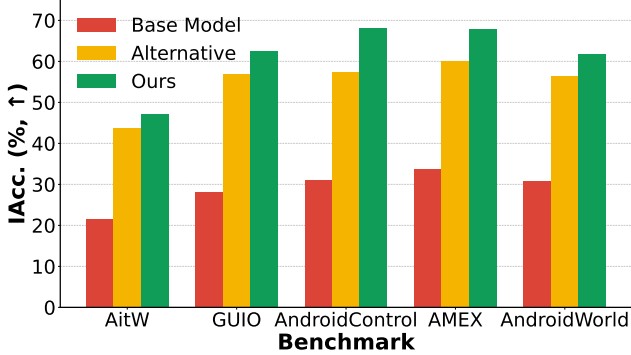

*Figure 6.* **Ablation on $R_t$ train data quality.** Our method consistently outperforms the naïve alternative in terms of IAcc. Both models are trained on 37K samples on top the base model `Qwen3 VL 8B`.

### 4.5. Further Analysis: Ablation

**Ablation Analysis on Generating $S_{t+1}^{\text{code}}$.** For our first ablative study, we examine whether the synthetic cross-modal policy trajectory repurposed training (see Step 1 and 2 in Fig. 3) is superior to a naïve alternative of generating next state code ($S_{t+1}^{\text{code}}$) with the same frontier model $S_{t+1}^{\text{code}} \leftarrow \pi^*(S_t^{\text{image}}, A_t, P^{\text{WM}})$, where $P^{\text{WM}}$ is the identical prompt used for all WM evaluations (see Appendix C). We randomly sample 100 $S_{t+1}^{\text{code}}$ instances (25 from each dataset) and evaluate them using our established metrics. Given the managable sample size, we employ the most capable (expensive) frontier models, `Gemini 3 Pro` and `Claude 4.5 Opus`, for our IAcc. evaluations. As presented in Tab. 3, our approach for generating $S_{t+1}^{\text{code}}$ outperforms the naïve alternative. Our approach generates renderable code 100% of the time, with a perfect 100% IAcc. score when using `Gemini 3 Pro` as the judge.

**Ablation Analysis on Generating $R_t$.** We evaluate the efficacy of our reasoning generation (Step 3 in Fig. 3) against a baseline $R_t^*$ generated by the same frontier model without look-ahead: $R_t^* \leftarrow \pi^*(S_t^{\text{image}}, A_t, P^{\text{WM}})$. We train two `gWorld` variants using `Qwen3 VL 8B` on 37K samples using the SFT dataset $\{(X : S_t^{\text{image}}, A_t, Y : \mathcal{R}_t, S_{t+1}^{code})\}_{t=1}^T$, $\mathcal{R}_t \in \{R_t, R_t^*\}$. Crucially, we use identical validated $S_{t+1}^{\text{code}}$ targets for both models to strictly isolate the performance impact of the reasoning trace ($R_t$ vs. $R_t^*$). As shown in Fig. 6, while both strategies improve world modeling, our method outperforms the alternative across all five benchmarks, con-

| Model | Avg. Rank ↓ | IAcc. (%) ↑ |
|---|---|---|
| gWorld 32B | **1.68** | **79.6** |
| gWorld 8B | 2.16 | 74.9 |
| Qwen3 VL 235B | 2.19 | 51.5 |
| GLM-4.6V 106B | 2.21 | 67.4 |
| Qwen3 VL 32B | 2.52 | 52.5 |
| Llama 4 402B | 2.58 | 55.7 |
| Qwen-I-E 20B | 2.73 | 13.4 |
| Emu3.5 34B | 2.77 | 25.8 |
| Llama 4 109B | 2.98 | 50.0 |
| Qwen3 VL 8B | 3.19 | 29.2 |

*Table 4.* **Human evaluation results.** We report average human preference rank, where lower is better, and IAcc. Human annotators rank anonymized candidate outputs given the current GUI state image, action, and ground-truth next state. The best values are **bolded** and the second best are underlined.

| Model | Photo-realistic (17.4%) | Others (81.5%) | Δ |
|---|---|---|---|
| gWorld 8B | **75.26** | 75.92 | -0.66 |
| gWorld 32B | 74.98 | **79.68** | -4.70 |
| GLM-4.6V 106B | 67.91 | 67.83 | +0.08 |
| Llama 4 402B | 54.35 | 54.80 | -0.45 |
| Qwen3 VL 235B | 50.05 | 48.03 | +2.02 |
| Qwen3 VL 32B | 49.19 | 52.92 | -3.73 |
| Qwen3 VL 8B | 25.02 | 29.34 | -4.32 |
| Emu3.5 34B | 22.92 | 24.90 | -1.98 |
| Qwen-I-E 20B | 14.04 | 12.56 | +1.48 |

*Table 5.* **Impact of photo-realistic content.** We report IAcc. (%) on photo-realistic transitions and other text/structure-heavy transitions. Δ denotes the absolute difference between photo-realistic and other transitions.

firming the superiority of our look-ahead strategy.

### 4.6. Further Analysis: Human Evaluation Study

We further conduct a human evaluation to validate our automatic VLM-as-a-Judge results. We randomly sample 100 examples, equal-weighted across all six benchmarks. For each sample, annotators are shown the current GUI state image, action, gold next state, and four anonymized candidate predictions in randomized order, then rank the outputs from best to worst. We recruit 12 annotators and collect three independent annotations per sample, yielding 300 total annotations.

As shown in Tab. 4, human judgments support our main conclusion: gWorld 32B and 8B rank first and second with a 1.68 and 2.16 average rank. We observe a high pairwise win rate of 77.5% and 61.4%, respectively, Additionally, the Spearman $\rho = 0.806$ and Kendall $\tau = 0.600$, suggesting that our automatic metric reliably reflects human preference.

### 4.7. Further Analysis: Photo-realistic States

We further analyze whether gWorld's code-based representation is limited by photo-realistic content, such as camera views. We classify all in-distribution test transitions using Gemini 3 Flash. Photo-realistic transitions constitute a minority of the benchmark (17.4%), while most transitions are text or structure-heavy (81.5%). The remaining 1.1% were classified as ambiguous. This suggests that the dominant challenge in mobile GUI world modeling is not pixel-level photo-realism, but accurately modeling semantic and structural GUI state changes.

As shown in Tab. 5, gWorld remains robust on photo-realistic transitions. gWorld 8B achieves 75.26% IAcc. on photo-realistic states and 75.92% on other states, showing only a 0.66% drop. gWorld 32B similarly achieves

74.98% on photo-realistic states and 79.68% on other states, while still substantially outperforming all baselines in both categories. These results indicate that localized losses in photo-realistic fidelity do not erase the gains from modeling GUI structure and action-conditioned state transitions.

### 4.8. Potential World Model-enhanced Policy Model Performance Gains

Finally, we demonstrate the downstream efficacy of our WM by applying it to enhance mobile GUI agent policies. We provide granular details in Appendix C.2.

**Experiment Set-up.** Inspired by Luo et al. (2025), we evaluate the potential performance gains achieved by integrating a WM into the policy via breadth-wise rollout and value estimation. Specifically, we present the M3A agent (Rawles et al., 2025) with $K = 3$ action candidates $\{A_t^1, \ldots, A_t^K\}$, which include the ground truth action. For the M3A + WM setting, we (1) roll out these $K$ candidate actions using the WM to predict next states $\{S_{t+1}^1, \ldots, S_{t+1}^K\}$, (2) compute the value of these transitions $\{V(S_t, A_t^k, S_{t+1}^k)\}_{k=1}^K$ by prompting the policy backbone, and (3) select the action with the highest estimated value. For the M3A + Value wo. WM baseline, the value function estimates utility directly from the current state and action, i.e., $\{V(S_t, A_t^k)\}_{k=1}^K$, without future state prediction. The value and policy models are set the same models to ensure that the method is self-contained.

**Results.** As shown in Tab. 6, across two arbitrarily set backbone policies (Gemini 2.5 Flash, GPT-5 Mini) with $K = 3$, incorporating gWorld 8B yields the most significant performance gains over the M3A baseline. Across gWorld 8B, Qwen3 VL 32B, and Qwen3 VL 8B, we observe that world modeling performance is positively correlated with downstream policy gains. On average, a 1.0 percentage point increase in world modeling performance translates to a 0.49 percentage point improvement in downstream policy.

| Method | GUIO | AC | AMEX | KApps | Avg. |
|---|---|---|---|---|---|
| Backbone Policy: | | Gemini 2.5 Flash | | | |
| M3A | 63.2 | 68.9 | 65.5 | 51.5 | 62.3 |
| + Value wo. WM | 54.4 | 58.5 | 53.9 | 43.1 | 52.5 |
| + Qwen3 VL 8B | 41.4 | 48.3 | 54.5 | 45.8 | 47.5 |
| + Qwen3 VL 32B | 57.4 | 70.9 | 70.5 | 53.6 | 63.1 |
| + gWorld 8B | **71.8** | **79.6** | **72.9** | **55.4** | **69.9** |
| Δ vs. Qwen3 VL 8B | +30.4 | +31.3 | +18.4 | +9.6 | +22.4 |
| Backbone Policy: | | GPT-5 Mini | | | |
| M3A | 42.8 | 59.5 | 49.7 | 25.0 | 44.2 |
| + Value wo. WM | 58.6 | 69.9 | 57.7 | 37.0 | 55.8 |
| + Qwen3 VL 8B | 42.8 | 47.9 | 51.8 | 39.2 | 45.4 |
| + Qwen3 VL 32B | 60.2 | 63.9 | 66.7 | 45.2 | 59.0 |
| + gWorld 8B | **72.4** | **75.2** | **74.1** | **47.0** | **67.2** |
| Δ vs. Qwen3 VL 8B | +29.6 | +27.3 | +22.3 | +7.8 | +21.8 |

*Table 6.* **Step-wise accuracy (%) comparison across world models.** Rows highlighted in gray leverage our proposed gWorld models. The Δ rows indicate the absolute performance gain over the corresponding Qwen3 VL baseline. The best scores are **bolded** and the second best are underlined.

## 5. Additional Discussion

**Evaluation Metric Comparison with VIMO.** While it was not possible to compare directly with VIMO on our experimental settings as they do not open-weight their diffusion model as of the time of writing, we compare Similarity v1 scores (in Appendix Tab. 12) on MWMBENCH-AITW and MWMBENCH-ANDROIDCONTROL as they are directly comparable with the experiments conducted in Luo et al. (2025). gWorld 8B and 32B achieves an average of 81%, 81.9%, while VIMO achieves 74% suggesting that the gWorld is superior in terms of generating next states closer to the gold example.

**Rendering Web Code is Virtually Overhead Free.** We empirically find that the wall-clock time of rendering the generated web code is virtually cost-free. When we start the experiment we incur a 1 second one-off wall-clock cost of launching the browser, but afterwards it takes approximated 0.3 seconds per render and capture. This minimal wall-clock time can also easily be parallelized using more process threads.

**End-to-End Inference Latency.** Beyond prediction quality, gWorld is substantially faster than prior pixel-based world models. Using vLLM (Kwon et al., 2023) on 4×H200 GPUs, gWorld 32B and 8B achieve approximately 5,000 and 20,000 tokens/sec, corresponding to generation latencies of 1.0s and 0.25s per state, respectively. Rendering adds only 0.3s per state after a one-time 1s browser initialization cost. Thus, the end-to-end latency is 1.3s and 0.55s, respectively. In contrast, VIMO reports 160s per state. Even if VIMO's reported 38s local model-execution time is removed by improving GPU compute, its remaining 122s sequential pipeline overhead is still 94× slower than gWorld 32B.

| Method | Latency (s) | Speed-up vs. VIMO |
|---|---|---|
| gWorld 8B | **0.55** | **291×** |
| gWorld 32B | 1.30 | 123× |
| VIMO | 160 | – |
| VIMO (idealized, −38s) | 122 | 1.31× |

*Table 7.* **End-to-end inference latency comparison.** We report per-state wall-clock latency. gWorld combines a single VLM generation pass with lightweight rendering, yielding 0.55s latency for the 8B model and 1.30s latency for the 32B model on 4×H200 GPUs. VIMO reports 160s end-to-end latency. Even after subtracting the reported 38s local model-execution time from VIMO, its remaining sequential pipeline overhead is 122s, which is still substantially slower than gWorld.

This highlights the practical benefit of generating directly renderable code rather than relying on a multi-stage OCR, masking, and diffusion pipeline.

**Potential Application: World Model for Synthetic Data Scaling and Scalable RL.** Mobile GUI agent training faces three key challenges that WMs can address. First, irreversible or consequential actions (e.g., financial transactions) are too risky to execute during training. Second, deep application states require many sequential actions to access, making data collection expensive. An accurate WM enables agents to simulate critical actions without real execution and to expand deep-state coverage by recursively generating trajectories from already-collected states. Third, online RL for GUI agents faces a fundamental scalability bottleneck due to *device-policy coupling*. Each rollout requires a persistent Android emulator, creating a 1:1 coupling where GPUs sit idle during action execution in emulators (>2s latency). WMs eliminate this device-bound bottleneck, enabling massively parallel, compute-bound rollout generation. We look forward to future works expanding on the wide-reaching applications of mobile GUI world models.

**Limitation and Future Direction.** We discuss the limitation and future direction of this work in Appendix E.

**Conclusion.** We presented gWorld, an open-weight, self-contained visual mobile GUI world model that predicts next states as renderable code. This representation shift preserves GUI structure and text fidelity while avoiding the complexity and latency of pixel-based pipelines. We further introduced MWMBENCH, a comprehensive benchmark for evaluating in-distribution, and OOD GUI transitions. Across MWMBENCH, gWorld establishes a new accuracy-model-size pareto frontier, scales predictably with more data, and improves downstream GUI policy performance. These results position renderable-code world modeling as a practical foundation for scalable and efficient mobile GUI agents.

## Acknowledgements

This research was fully funded by Trillion Labs. We acknowledge the following members of Trillion Labs who participated in collecting the data for MWMBENCH-KAPPS: Hongjoon Ahn, Hyungguk Kim, Juyoung Suk, Kyuseok Kim, Suyeoung An, and Wonsuk Yang. Special thanks to Hongjoon Ahn and Juyoung Suk for their valuable discussions. We would also like to thank Haein Lee for their assistance in designing Figures 2, 3, 13, 14, and 15. We also thank Hyungguk Kim, Hyunjin Seo, Joonkee Kim, Jungwoo Lee, Juyoung Suk, Kyuhee Jo, Minchan Jeong, Sangwon Jung, Suyeong An, Wonsuk Yang, and Yujin Kim for participating in the human evaluation study during the rebuttal phase.

## Impact Statement

This paper presents work whose goal is to advance the field of Machine Learning by enabling more capable and efficient mobile agents and world models. The primary societal benefits include enhancing digital accessibility for users with impairments and democratizing AI research through high-performance open-weight (as of publication) models that require less compute than prior methods. While improved GUI automation carries dual-use risks (e.g., automated fraud), we believe open research is crucial for developing robust safety measures.

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

# A. Further Details on our Method

The pseudocode of our method is available in Alg. 1. Prompt $P^{\text{img-to-code}}$ and $P^{\text{look-ahead}}$ is available below.

---

**Algorithm 1:** `gWorld`: World Model SFT Data Generation

---

**Input** : Offline policy trajectories $\mathcal{D}^\pi = \{\tau\}$, where $\tau = \{(S_t^{\text{image}}, A_t)\}_{t=1}^T$; frontier model $\pi^*$; prompts $P^{\text{img-to-code}}$ and $P^{\text{look-ahead}}$

**Output:** World-model SFT dataset $\mathcal{D}^{\text{WM}} = \{(X : (S_t^{\text{image}}, A_t),\ Y : (R_t, S_{t+1}^{\text{code}}))\}$

```
/* Initialize output dataset                                                      */
```
$\mathcal{D}^{\text{WM}} \leftarrow [];$

**for each** episode $\tau \in \mathcal{D}^\pi$ **do**

    Extract $\{(S_t^{\text{image}}, A_t)\}_{t=1}^T$ from $\tau$;

```
    /* (1) Repurpose policy trajectory:  iterate transitions St → St+1         */
```
    **for each** timestep $t \in \{1, \ldots, T-1\}$ **do**

```
        /* (2) Synthetic cross-modal re-labeling of next-state St+1            */
```
        $S_{t+1}^{\text{code}} \leftarrow \pi^*(S_{t+1}^{\text{image}}, P^{\text{img-to-code}});$

```
        /* (3) Reasoning generation with free look-ahead to St+1              */
```
        $R_t \leftarrow \pi^*(S_t^{\text{image}}, A_t, S_{t+1}^{\text{image}}, P^{\text{look-ahead}});$

```
        /* Construct SFT pair                                                   */
```
        $X \leftarrow (S_t^{\text{image}}, A_t);$

        $Y \leftarrow (R_t, S_{t+1}^{\text{code}});$

        $\mathcal{D}^{\text{WM}}.\text{append}((X, Y));$

**return** $\mathcal{D}^{WM}$

---

---

Prompt $P^{\text{img-to-code}}$: Mobile GUI Image to Code

You are an expert mobile UI developer. Given a screenshot of a mobile interface, you must first analyze it and then generate clean, responsive HTML code.

Your task has TWO steps:
1. REASONING: Analyze the screenshot and plan the HTML structure.
2. HTML GENERATION: Create the HTML code based on your analysis.

Focus on these two critical criteria:
1. Each button's function should be "inferable" / "differentiable" - users must be able to understand what each button does.
2. Each text content should be well-represented in the HTML output - all visible text must be accurately captured.

In your REASONING, address:
- The overall structure and layout of the screen (header, main content, footer, etc.)
- Important UI elements and their hierarchy (buttons, text, images, icons, etc.)
- Which parts of the screen are most important for functionality
- How to ensure buttons are clearly differentiated and their functions are inferable
- How to accurately represent all text content
- Color scheme and visual styling that supports clarity
- Any interactive elements and their purposes

Requirements for HTML:
1. Generate complete, valid HTML5 code.
2. Choose between using inline CSS and utility classes from Bootstrap, Tailwind CSS, or MUI for styling.
3. Use mobile-first design principles matching screenshot dimensions.
4. For images, use inline SVG placeholders with explicit width and height.
5. Make it visually as close to the provided screenshot.
6. Each button's function must be "inferable" / "differentiable".
7. All text content from the screenshot must be well-represented.

Return ONLY a JSON object with this exact structure:
{
    "reasoning": "Your detailed analysis and planning here",
    "html": "Your complete HTML code here"
}

| Split | #Trans | #Apps | Lang | Example Apps |
|---|---|---|---|---|
| ANDROIDWORLD | 686 | 18 | EN | **Productivity:** Joplin, Markor, Tasks, Simple Calendar
**Media:** Retro Music, Simple Gallery
**Navigation:** OpenTracks |
| KAPPS | 495 | 14 | KO | **Food & Shopping:** Baemin, Coupang, Coupang Eats
**Mobility:** Naver Map, Uber
**Communication:** KakaoTalk, Discord, Gmail |

*Table 8.* **Composition of MWMBENCH's out-of-distribution splits.** AndroidWorld primarily consists of open-source productivity apps, while KApps features popular Korean apps with Korean-language interfaces. Both splits cover apps and domains that are underrepresented in our training data.

---

**Prompt $P^{\text{look-ahead}}$: Look-Ahead Reasoning Synthesis**

You are a GUI Agent.
Action: {action}

You are given a current screenshot state (first image), action and the next state (second image).

Action is also visually annotated in the first image
1. Clicks: Red circle with crosshair + yellow center dot
2. Scrolls: Blue line with green start point + red end point or based on direction

Generate reasoning on what this next state would look like as if you were only given the current screenshot. Focus only on the changes that can be predicted from the current screenshots. In the reasoning, do not mention the visual annotation of the action or the existence of the ground truth next state. Only generate the reasoning, nothing else.

---

## B. Extended Details on MWMBench

The composition of our OOD splits is summarized in Tab. 8, and the detailed distribution of transitions across apps and domains is shown in Fig. 7.

### B.1. MWMBENCH-ANDROIDWORLD

MWMBENCH-ANDROIDWORLD was collected by running M3A (Rawles et al., 2025) with `Qwen3 VL 235B-A22B` as the base policy model on AndroidWorld. The resulting benchmark contains 686 transitions across 88 episodes spanning 18 distinct applications.

**Deduplication.** The M3A agent often undergoes extensive trial-and-error during task execution, repeatedly visiting similar states or performing redundant actions before reaching the goal. This results in trajectories with numerous nearly identical transitions. If used directly for evaluation, such redundancy would cause the benchmark to disproportionately weight certain transitions, skewing the assessment of world model performance. To ensure a balanced evaluation across diverse situations, we apply a two-stage deduplication procedure to remove redundant transitions.

In the first stage, we automatically identify candidate duplicates based on visual similarity. We group transitions by their (app, action) pairs and compute pairwise similarity within each group by comparing both the pre-action screenshot $S_t$ and post-action screenshot $S_{t+1}$. Transitions with both similarities exceeding 0.997 are clustered via connected components, producing candidate duplicate groups.

In the second stage, human annotators manually review each candidate cluster to verify whether the transitions are truly redundant. Only transitions confirmed as duplicates through this manual inspection are removed, with one representative retained per cluster.

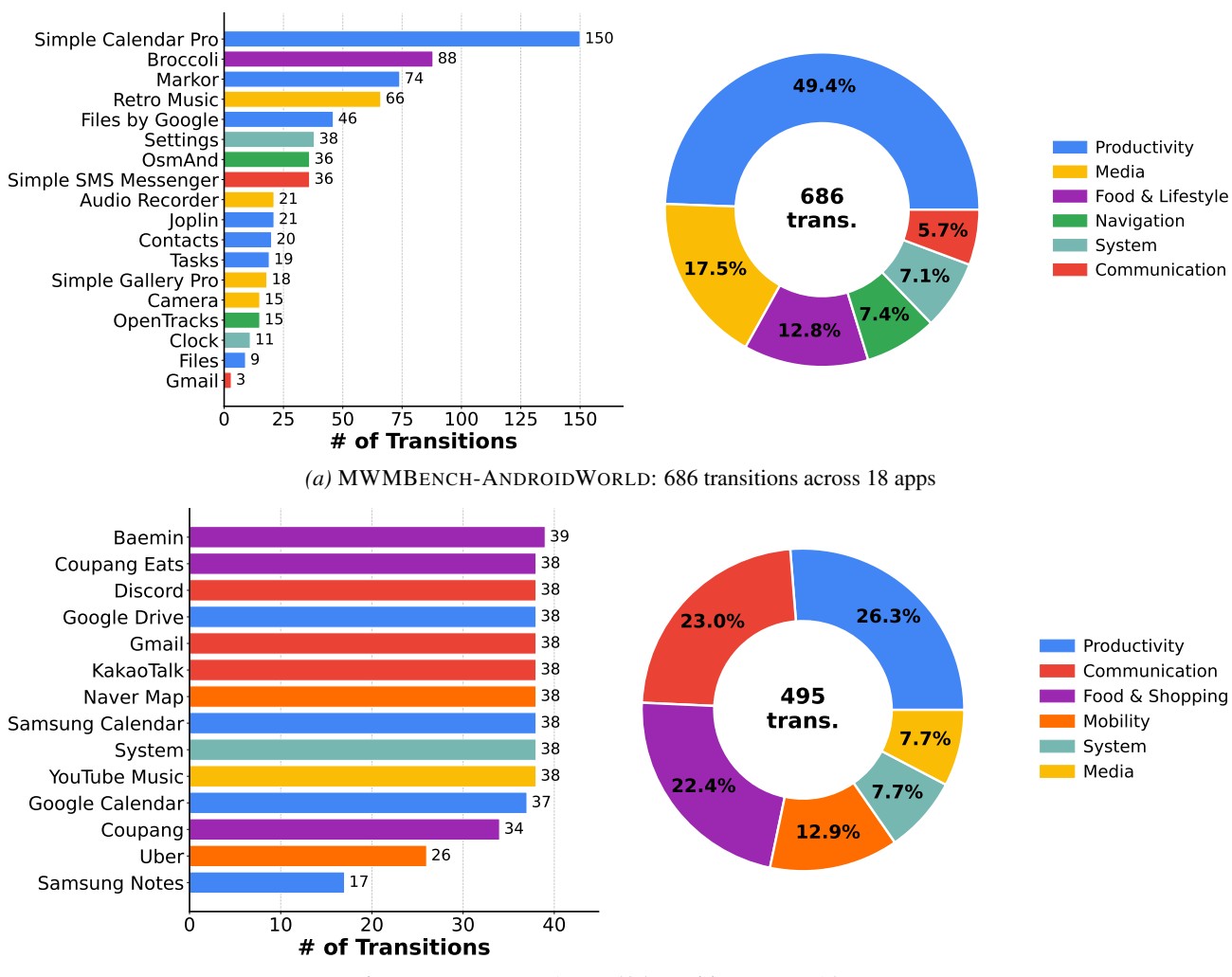

*(a)* MWMBENCH-ANDROIDWORLD: 686 transitions across 18 apps

*(b)* MWMBENCH-KAPPS: 495 transitions across 14 apps

*Figure 7.* **Distribution of transitions in MWMBench's OOD splits.** Left: per-app transition counts (colors indicate domain). Right: domain-wise distribution. AndroidWorld is dominated by productivity apps (55.4%), while KApps shows a more balanced distribution across food & shopping, communication, and productivity domains.

This process reduces the dataset from 1,094 to 686 transitions (37% reduction), removing redundant transitions such as repeated app launches and common navigation patterns while preserving task-specific unique transitions.

### B.2. MWMBENCH-KAPPS

MWMBench-KApps was collected manually in-house by technical staff members selected for their proficiency in Korean and English. The statistics for the top 10 most popular apps in Korea were used to determine which apps to collect data from. Both virtual and physical devices were used for collection, as some target apps were not supported on emulators. The collection interface was built using a fixed overlay that first records actions, then saves screenshots and accessibility trees. After the state is saved, interface converts the action into a Android Virtual Device (AVD) command for execution. For accessibility tree communication, gRPC and HTTP were used for virtual and physical devices, respectively. However, for this work, we only utilize the screenshots and actions. Figure 8 shows the collection interface, and Table 9 shows the action space used for collection. After collection, each episode was manually filtered for action quality, excluding episodes that did not properly progress toward the goal at every step.

**Out-of-Distribution Setting.** MWMBENCH-KAPPS serves as an OOD evaluation set for assessing multilingual generalization capabilities. While our training data (AITW, AndroidControl, AMEX, and GUI Odyssey) and most existing

*Table 9.* Action space used for Kapps data collection.

| Action | Parameters |
|---|---|
| click | [x, y] |
| swipe | [start_x, start_y, velocity, end_x, end_y] |
| system_button | {recent, home, back} |
| set_text | [x, y], text |
| long_press | [x, y] |
| wait | duration |
| complete | comment (optional) |
| impossible | comment (optional) |
| launch_app | package_name |

world model research are predominantly English-centric, MWMBENCH-KAPPS is entirely Korean-based: all task goals are written in Korean, and 94.5% of transitions contain Korean text in their UI screenshots. Additionally, KApps features popular Korean applications (e.g., Baemin, Coupang, KakaoTalk, Naver Map) that are absent from our English-focused training data. This benchmark thus provides a unique testbed for evaluating whether world models can generalize beyond their monolingual training distribution to unseen languages and regional applications.

## C. Further Details on Experiments

**Hardware.** Experiments were conducted on a cluster of up to four H200 nodes. Each node comprises eight NVIDIA H200 GPUs, featuring intra-node communication via NVLink and inter-node connectivity through InfiniBand.

**Training Settings.** We used the same hyperparameters for training both `Qwen3 VL 8B` and `Qwen3 VL 32B`. See Tab. 10 for the complete set of hyperparameters. We utilized the training code for finetuning made available by the Qwen team: https://github.com/QwenLM/Qwen3-VL/tree/main/qwen-vl-finetune. We only train the LLM and MLP projector layer. We freeze the vision encoder as unfreezing did not lead to a meaningful improvement in performance.

*Table 10.* Training Hyperparameters

| Hyperparameter | Value |
|---|---|
| Batch Size | 64 |
| Learning Rate | 2e-7 |
| MLP Learning Rate | 2e-7 |
| LR Scheduler | Cosine |
| Warm up Ratio | 0.01 |
| Weight Decay | 0.01 |
| Training Epochs | 5 |
| Max Image Pixels | 4,233,600 |
| Min Image Pixels | 3,136 |
| *Trainable Components* | |
| Vision Encoder | Frozen |
| MLP Projector | Tuned |
| LLM | Tuned |

**Evaluation and Inference Settings.** We use greedy decoding (temperature = 0) and set the max model length to 16384 such that there is no premature cut-off.

**World Model Evaluation Prompt.** We use the same $P^{\text{WM}}$ prompt for all models. This prompt was first curated based on zero-shot performance on `Qwen3 VL 235B-A22B`. We apply model-specific chat templates to ensure consistent formatting across different architectures.

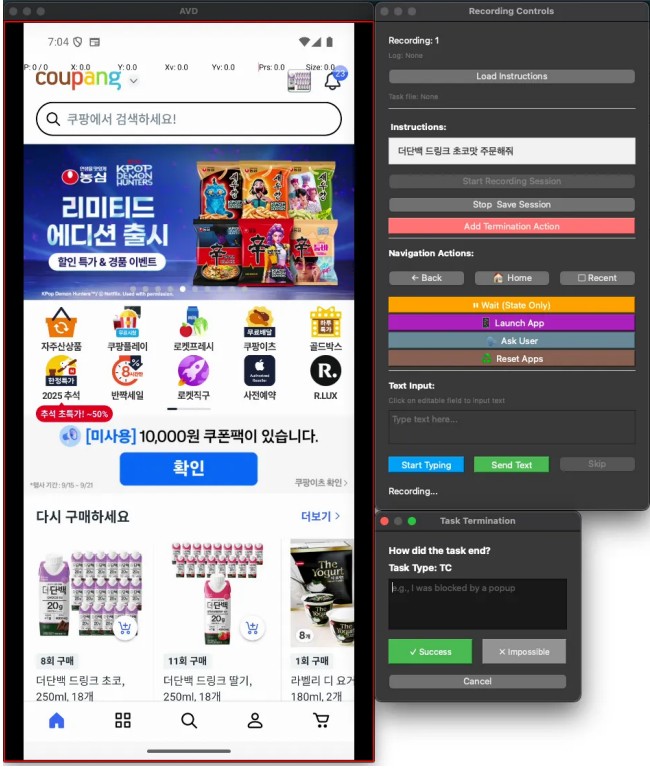

*Figure 8.* Data collection interface software built for KApps.

---

**Prompt $P^{\text{WM}}$: World Model Evaluation**

You are an expert mobile UI World Model that can accurately predict the next state given an action. Given a screenshot of a mobile interface and an action, you must generate clean, responsive HTML code that represents the state of the interface AFTER the action is performed. First generate reasoning about what the next state should look like based on the action. Afterwards, generate the HTML code representing the next state that logically follows the action. You will render this HTML in a mobile viewport to see how similar it looks and acts like the mobile screenshot.

**Requirements:**
1. Provide reasoning about what the next state should look like based on the action
2. Generate complete, valid HTML5 code
3. Choose between using inline CSS and utility classes from Bootstrap, Tailwind CSS, or MUI for styling, depending on which option generates the closest code to the screenshot.
4. Use mobile-first design principles matching screenshot dimensions.
5. For images, use inline SVG placeholders with explicit width and height attributes that match the approximate dimensions from the screenshot. Matching the approximate color is also good.
6. Use modern web standards and best practices
7. Return ONLY the HTML code, no explanations or markdown formatting
8. The generated HTML should render properly in a mobile viewport.
9. Generated HTML should look like the screen that logically follows the current screen and the action.

Action: {action}

Output format:
Next State Reasoning: <your reasoning about what the next state should look like>
HTML: <valid_html_code >

Generate the next state reasoning and the next state in html:

**Metric: Instruction Accuracy.** Instruction Accuracy is obtained by: IAcc. $\leftarrow \pi^*(S_t, A_t, \hat{S}_{t+1}, P^{\text{IAcc.}})$ where $\hat{S}_{t+1}$ is generated by our model of interest, and the prompt $P^{\text{IAcc.}}$ follows Luo et al. (2025).

---

Prompt $P^{\text{IAcc.}}$: VLM-as-a-Judge Evaluation of Generated Next State

You are an expert in evaluating the performance of a mobile emulator. The mobile emulator is designed to navigate the UI change based on human instruction.

Inputs:
Current UI Screenshot: The present state of the cellphone's user interface.
Next UI Screenshot: The mobile emulator generated UI indicating the next state of the cellphone's user interface based on human instruction.
Human instruction: The action applied on the current UI screenshot.

Your goal is to determine whether the mobile emulator successfully predicts the next UI image with current information and layout based on the current UI and the user action.

Consider these aspects:
- Does the generated UI show a plausible result of applying the action?
- Is the layout and structure consistent with what would happen after the action?
- Are interactive elements (buttons, inputs, etc.) in expected states?
- Does the content reflect the expected changes from the action?

**IMPORTANT**
Format your response into a JSON map as shown below:
{
    "Thoughts": "<your thoughts and reasoning process>",
    "Status": "success" or "failure"
}

---

### C.1. Further Details on Scaling Analysis

While we only generate up to 240K samples for training, Tab. 11 reports a maximum of 3.7 million samples available for training.

| Dataset | Episodes | Policy Transitions | Available World Model Transitions |
|---|---|---|---|
| GUIOdyssey | 8,334 | 119,559 | 111,225 |
| Android Control | 14,501 | 73,968 | 59,467 |
| AitW | 707,186 | 4,232,911 | 3,525,725 |
| AMEX | 3,046 | 35,661 | 32,615 |
| **Total** | **733,067** | **4,462,099** | **3,729,032** |

*Table 11.* Maximum existing transitions available for training `gWorld`

### C.2. Further Details on World Model-enhanced Policy Experiments

We implement an *oracle* variant of M3A policy agent (Rawles et al., 2025) so that we can clearly observe potential gains via world modeling. We provide the agent with the ground truth action, current screenshot $S_t$, goal $G$, and history of natural language actions $H_t$. Given the ground truth action $A_t^{GT}$, it first generates $K-1$ alternatives (see $P^{\text{alt}}$). The agent then selects the *best* action from all $K$ candidates to progress toward the goal (see $P^{\text{select}}$). Accuracy measures how often the agent selects the ground truth. We formalize this procedure in Algorithm 2. This setup isolates the policy's selection ability from its generation ability in a single-step evaluation. We adopt this setting as most policies we tested failed to show meaningful improvements at higher $K$ i.e., often failed to generate at least one correct action among $K$ candidates. We note

that enabling the policy for effective test-time scaling remains an open challenge and is beyond the scope of this work. Here, we focus on quantifying the improvement gains from using a world model as a value function.

The next baseline augments M3A with a value function without the world model. After generating $K - 1$ alternatives, each action including the ground truth is passed to the backbone policy model in parallel to judge its validity and assign a confidence score (see $P^{\text{value-no-wm}}$). The valid action with the highest confidence is selected. We outline this in Algorithm 3.

Finally, M3A augmented with a WM which generates the next state for each of the $K$ actions. Each next state is provided along with its corresponding action to the backbone policy model in parallel to judge validity and assign a confidence score (see $P^{\text{value-wm}}$). The highest-scored valid action is selected. The full procedure is given in Algorithm 4.

---

**Prompt $P^{\text{alt}}$: M3A Alternative Action Generation**

You are an AI agent that can operate an Android phone. Given a goal and the current screenshot, suggest alternative actions that could be taken.
Goal: {goal}

Previous actions: {history}
The following action has already been suggested (DO NOT repeat this action): {gt_action}
Available action types:

- TAP: Tap on a location. Format: {{"action_type": "TAP", "x": <x>, "y": <y>}}
- SCROLL: Scroll in a direction. Format: {{"action_type": "SCROLL", "direction": "<up|down|left|right>"}}
- TYPE: Type text. Format: {{"action_type": "TYPE", "text": "<text>"}}
- BACK: Press back button. Format: {{"action_type": "BACK"}}
- HOME: Press home button. Format: {{"action_type": "HOME"}}
- ENTER: Press enter key. Format: {{"action_type": "ENTER"}}
- LONG_PRESS: Long press on a location. Format: {{"action_type": "LONG_PRESS", "x": <x>, "y": <y>}}

Coordinates are in range [0, 1000] where (0,0) is top-left and (1000,1000) is bottom-right.
Suggest {num_alternatives} DIFFERENT alternative actions that could also make progress toward the goal.

Critical requirements:

1. Each alternative MUST be a completely different action from the one already suggested above.
2. Do NOT repeat or slightly modify the already-suggested action (e.g., if the suggested action is a TAP at (500, 300), do NOT suggest a TAP at (500, 301) or nearby coordinates).
3. For TAP actions, choose DIFFERENT UI elements to tap, not the same element with slightly different coordinates.
4. Each alternative should represent a meaningfully different approach to achieving the goal.

For each action, explain the reasoning behind it.
You must output exactly {num_alternatives} actions numbered 1 to {num_alternatives}:

{{1: {{Reason: ..., Action: {{"action_type":...}}}}, ..., {num_alternatives}: {{Reason: ..., Action: {{"action_type":...}}}}}}

---

**Prompt $P^{\text{select}}$: Action Selection from $K$ Action Candidates**

You are an AI agent that can operate an Android phone. Given a goal and the current screenshot, select the best action from the candidates below.
Goal: {goal}

Previous actions: {history}
Candidate actions: {candidates}
Available action types:

- TAP: Tap on a location. Format: {{"action_type": "TAP", "x": <x>, "y": <y>}}
- SCROLL: Scroll in a direction. Format: {{"action_type": "SCROLL", "direction": "<up|down|left|right>"}}
- TYPE: Type text. Format: {{"action_type": "TYPE", "text": "<text>"}}
- BACK: Press back button. Format: {{"action_type": "BACK"}}
- HOME: Press home button. Format: {{"action_type": "HOME"}}
- ENTER: Press enter key. Format: {{"action_type": "ENTER"}}
- LONG_PRESS: Long press on a location. Format: {{"action_type": "LONG_PRESS", "x": <x>, "y": <y>}}

Coordinates are in range [0, 1000] where (0,0) is top-left and (1000,1000) is bottom-right.
Analyze each candidate action carefully based on the screenshot and goal. Select the candidate most likely to help

achieve the goal.
Output format:

Reason: <your analysis of why this candidate is best>
Best: <candidate number>

---

**Prompt $P^{\text{value-no-wm}}$: Value Estimation without World Model**

You are an AI agent evaluating whether an action will help achieve a goal on an Android phone.
Goal: {goal}

Previous actions: {history}
The action being evaluated: {action}
Reason for this action: {reason}
You are given the CURRENT screenshot showing the UI state before the action. Available action types:

- TAP: Tap on a location. Format: {{"action_type": "TAP", "x": <x>, "y": <y>}}
- SCROLL: Scroll in a direction. Format: {{"action_type": "SCROLL", "direction": "<up|down|left|right>"}}
- TYPE: Type text. Format: {{"action_type": "TYPE", "text": "<text>"}}
- BACK: Press back button. Format: {{"action_type": "BACK"}}
- HOME: Press home button. Format: {{"action_type": "HOME"}}
- ENTER: Press enter key. Format: {{"action_type": "ENTER"}}
- LONG_PRESS: Long press on a location. Format: {{"action_type": "LONG_PRESS", "x": <x>, "y": <y>}}
- OPEN_APP: Open an app. Format: {{"action_type": "OPEN_APP", "app_name": "<name>"}}

Coordinates are in range [0, 1000] where (0,0) is top-left and (1000,1000) is bottom-right.
Your task is to judge whether the action is a reasonable step toward achieving the goal based on the current UI state.

Evaluate based on these criteria:

1. Does the action target the correct UI element or area visible on screen?
2. Is the action type appropriate for the current context?
3. How directly does this action advance the goal vs. being a roundabout step?

Respond in JSON format:
{{"Reason": "Your explanation", "Judgement": "valid" or "invalid", "Confidence": <score>}}
IMPORTANT: Use the FULL range of confidence scores to differentiate action quality:

- 0.9–1.0: Clearly the optimal action, directly advances the goal
- 0.7–0.8: Good action, makes progress but may not be the most efficient path
- 0.5–0.6: Acceptable action, loosely related to goal but indirect
- 0.3–0.4: Weak action, unlikely to help but not harmful
- 0.1–0.2: Poor action, probably wrong target or type

Avoid defaulting to 1.0 or 0.9 unless the action is clearly optimal. Be critical and discriminating.

---

**Prompt $P^{\text{value-wm}}$: Value Estimation With World Model**

You are an AI agent evaluating whether a predicted action will help achieve a goal on an Android phone.
Goal: {goal}

Previous actions: {history}
The action being evaluated: {action}
Reason for this action: {reason}
You will be given two screenshots:

1. BEFORE screenshot: The current UI state before the action
2. AFTER screenshot: The predicted UI state after performing the action

Available action types:
• TAP: Tap on a location. Format: {{"action_type": "TAP", "x": <x>, "y": <y>}}
• SCROLL: Scroll in a direction. Format: {{"action_type": "SCROLL", "direction": "<up|down|left|right>"}}
• TYPE: Type text. Format: {{"action_type": "TYPE", "text": "<text>"}}
• BACK: Press back button. Format: {{"action_type": "BACK"}}
• HOME: Press home button. Format: {{"action_type": "HOME"}}
• ENTER: Press enter key. Format: {{"action_type": "ENTER"}}
• LONG_PRESS: Long press on a location. Format: {{"action_type": "LONG_PRESS", "x": <x>, "y": <y>}}
• OPEN_APP: Open an app. Format: {{"action_type": "OPEN_APP", "app_name": "<name>"}}

Coordinates are in range [0, 1000] where (0,0) is top-left and (1000,1000) is bottom-right.
Your task is to judge whether the action is a reasonable step toward achieving the goal.

Evaluate based on this criterion: Does the predicted "after" screenshot show expected progress toward the goal?

Respond in JSON format:

{{"Reason": "Your explanation", "Judgement": "valid" or "invalid", "Confidence": <score>}}
IMPORTANT: Use the FULL range of confidence scores to differentiate action quality:

• 0.9–1.0: Clearly the optimal action, directly advances the goal
• 0.7–0.8: Good action, makes progress but may not be the most efficient path
• 0.5–0.6: Acceptable action, loosely related to goal but indirect
• 0.3–0.4: Weak action, unlikely to help but not harmful
• 0.1–0.2: Poor action, probably wrong target or type

Avoid defaulting to 1.0 or 0.9 unless the action is clearly optimal. Be critical and discriminating.

## D. Extended World Modeling Experiment Results

Tab. 12 organizes results for each of the different VLM-as-a-Judges, and each of the vision encoders. Training curves and data scaling analysis is presented in Fig. 9, 10. High inter-judge rank correlation is visualized in Fig. 11 and 12. Additional qualitative results are available in Fig. 13, 14, 15, 16, 17, 18, 19, 20. Fig. 21 is Fig. 4 with `Qwen-Image-Edit 20B`.

## E. Limitations and Future Work

While `gWorld` establishes a new paradigm for visual world modeling, we identify several limitations inherent to the current approach that pave the way for future research.

First, regarding data scale, we currently utilize only 260K training samples out of a potential pool of 3.7 million transitions (approximately 7% of available data). Given the predictable power-law scaling demonstrated in Figure 5, future work can

**Algorithm 2:** M3A: Oracle Policy Evaluation

**Input** : Test samples $\mathcal{D}^{\text{test}} = \{(S_t, A_t^{\text{GT}}, G, H_t)\}$ where $S_t$ is the current screenshot, $A_t^{\text{GT}}$ is the ground truth action, $G$ is the goal, and $H_t$ is the action history; policy model $\pi$; number of candidates $K$; prompts $P^{\text{alt}}$, $P^{\text{select}}$

**Output** : Accuracy (rate of selecting ground truth)

correct $\leftarrow 0$;

**for** *each sample* $(S_t, A_t^{\text{GT}}, G, H_t) \in \mathcal{D}^{\text{test}}$ **do**

```
/* (1) Build candidate set with GT as first candidate          */
```
$\quad \mathcal{C} \leftarrow \{1 : A_t^{\text{GT}}\}$;
```
/* (2) Generate K − 1 alternative actions                      */
```
$\quad \mathcal{A}^{\text{alt}} \leftarrow \pi(S_t, G, H_t, A_t^{\text{GT}}, P^{\text{alt}})$;

**for** $i \leftarrow 2$ **to** $K$ **do**
$\quad\quad \mathcal{C}[i] \leftarrow \mathcal{A}^{\text{alt}}[i-1]$;
```
/* (3) Policy selects best action from candidates              */
```
$\quad A_t^{\text{selected}} \leftarrow \pi(S_t, G, H_t, \mathcal{C}, P^{\text{select}})$;
```
/* (4) Check if selected action is ground truth                */
```
**if** $A_t^{selected} = A_t^{GT}$ **then**
$\quad\quad$ correct $\leftarrow$ correct $+ 1$;

**return** $correct/|\mathcal{D}^{test}|$

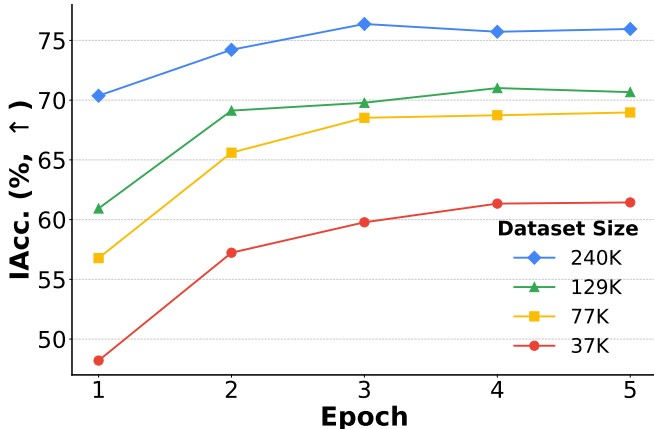

*Figure 9.* **Data scaling analysis.** We report average Instruction Accuracy (IAcc.) across the four in-distribution test splits as we scale the repurposed training data from 37K to 240K examples. The results demonstrate **strong positive scaling**.

significantly enhance performance by scaling the training data to utilize the full set of available offline trajectories.

Second, there are fundamental limitations to rendering photo-realistic content via web code. For instance, a video player displaying complex natural imagery is difficult to reconstruct with high fidelity under the current paradigm. While we posit that this does not significantly impact the semantic utility of mobile GUI world modeling or downstream policy performance, future work may explore hybrid techniques to address these specific visual failure cases.

Finally, we aim to extend gWorld beyond the single-frame Markov assumption by incorporating explicit working memory. While our current model demonstrates robust recursive prediction, many GUI environments exhibit long-range temporal dependencies that require context from previous interactions (e.g., maintaining state for a shopping basket across multiple pages). Transitioning from a single-observation state to a memory-augmented framework will be essential for capturing these long-term dependencies, ultimately developing gWorld into a realistic simulator capable of supporting long-horizon online reinforcement learning.

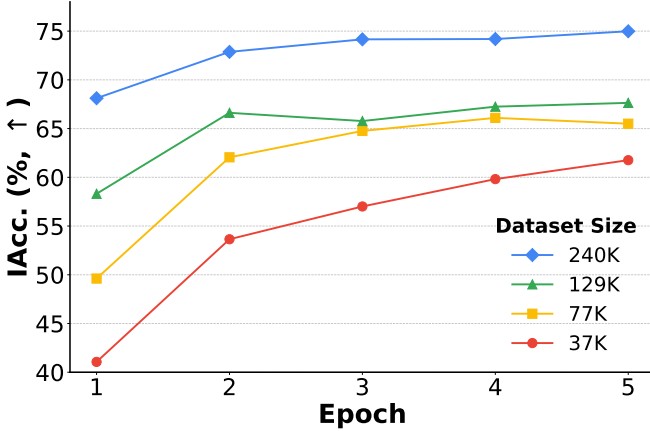

*Figure 10.* **Data scaling analysis.** We report average Instruction Accuracy (IAcc.) in MWMBENCH-ANDROIDWORLD. as we scale the repurposed training data from 37K to 240K examples. The results demonstrate **strong positive scaling**.

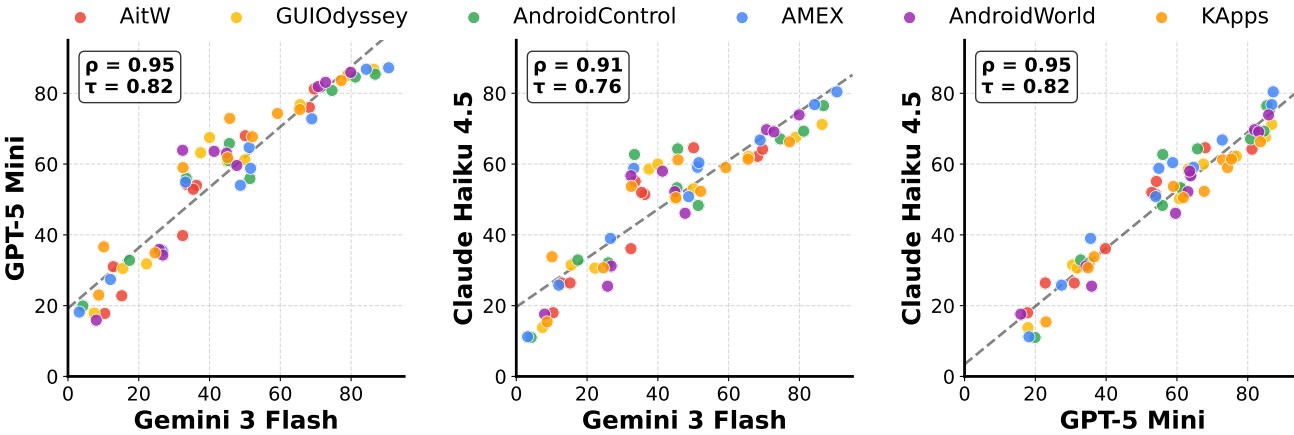

*Figure 11.* **Inter-judge agreement on world model instruction-following.** Each scatter plot compares Instruction-following Accuracy (IAcc., %) scored by two different VLM-as-a-Judge models (Gemini 3 Flash, GPT-5 Mini, Claude Haiku 4.5) across MWMBENCH datasets (AitW, GUIOdyssey, AndroidControl, AMEX, AndroidWorld, KApps). Each point corresponds to a (WM model, dataset) result, colors denote datasets, and the dashed line shows the linear regression fit. Spearman's $\rho$ and Kendall's $\tau$ show strong rank and score consistency across judges.

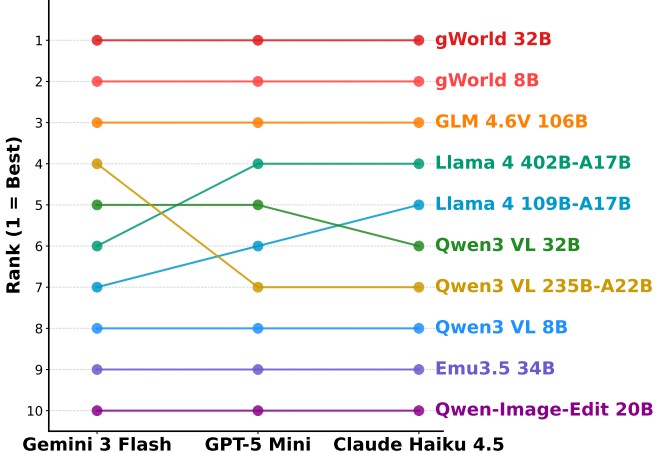

*Figure 12.* **Consistency of model rankings across VLM-as-a-Judge choices.** Bump chart shows the relative rank (1 = best) of each world model under different judges (`Gemini 3 Flash`, `GPT-5 Mini`, `Claude Haiku 4.5`), where ranks are determined by average IAcc. over MWMBENCH datasets. Rankings remain largely stable across judges, with `gWorld` consistently achieving top performance compared to other code-generation and image-generation baselines.

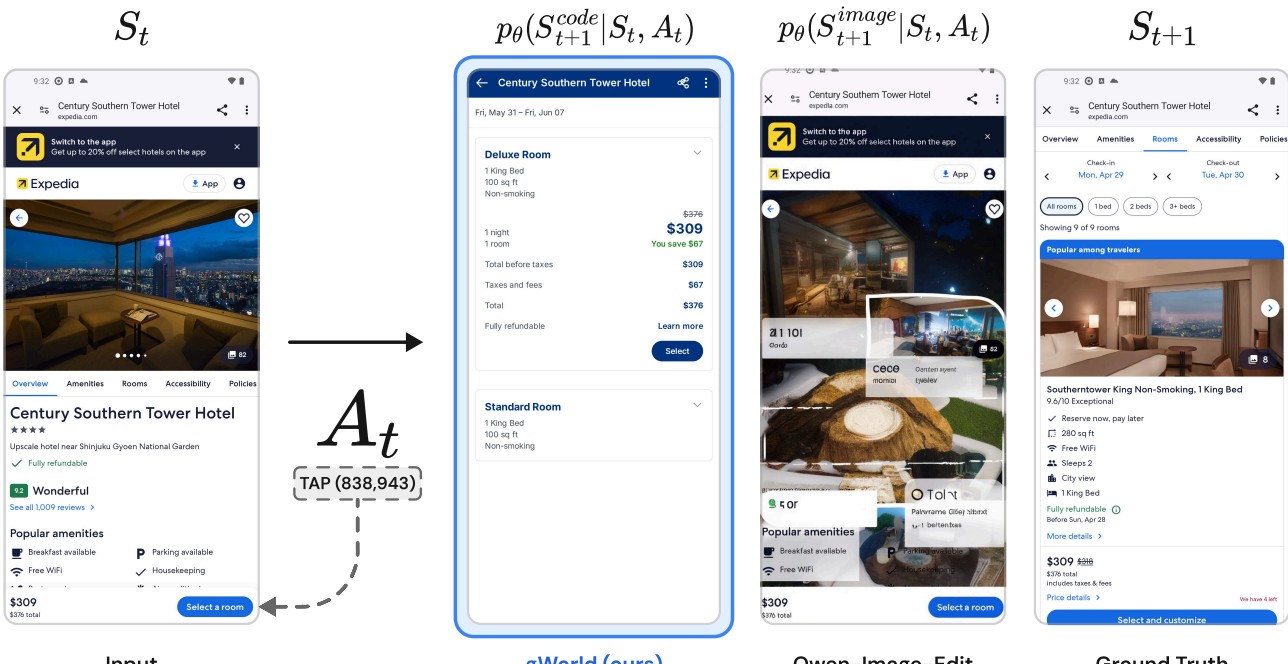

Figure 13. **Additional qualitative example 1.**

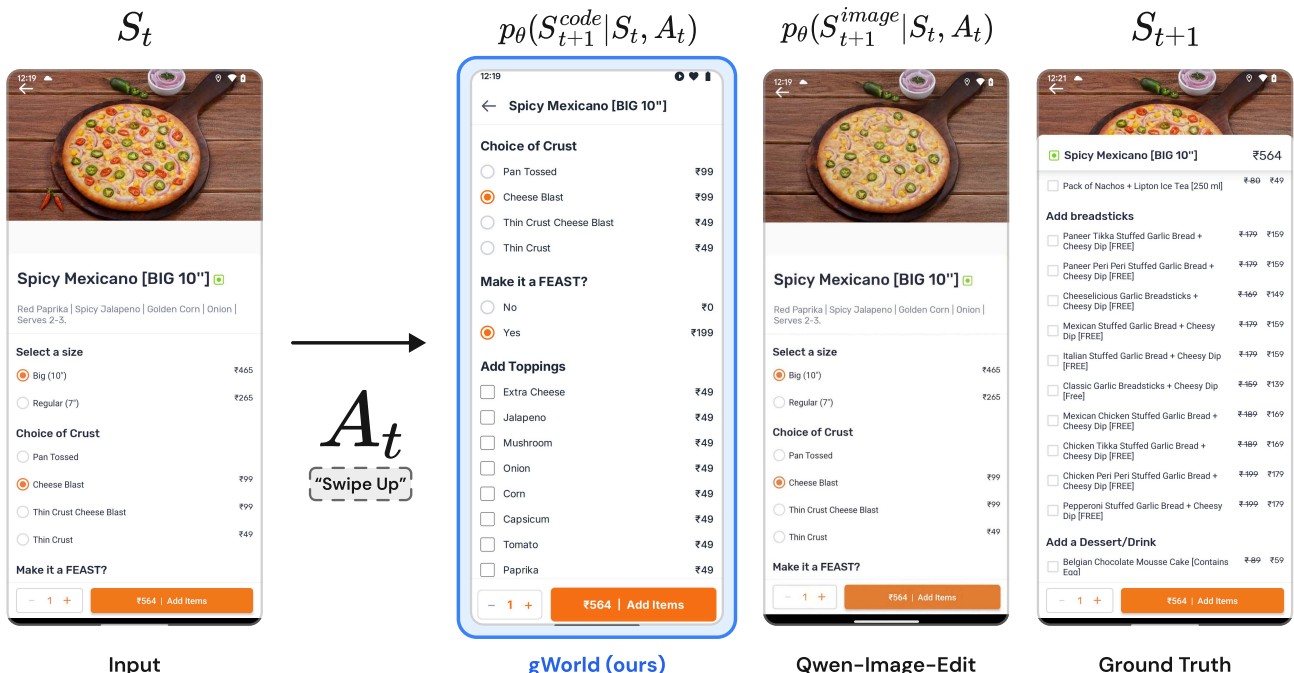

Figure 14. **Additional qualitative example 2.**

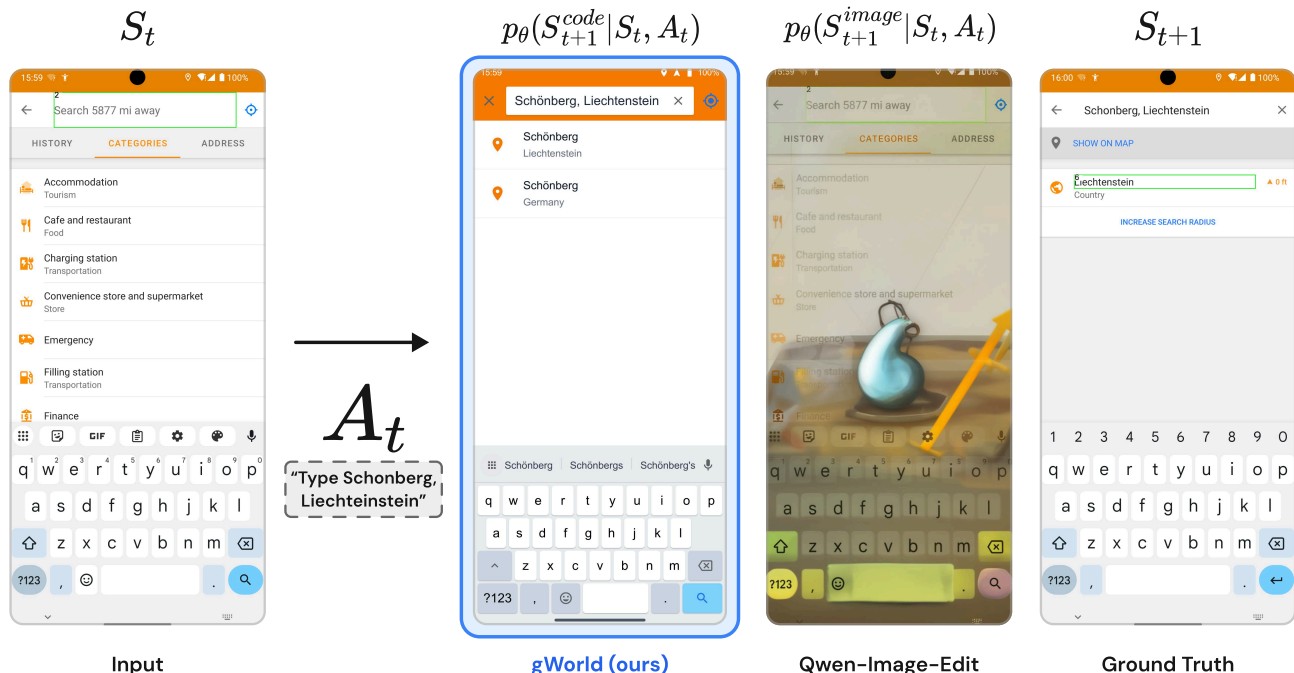

*Figure 15.* **Additional qualitative example 3.**

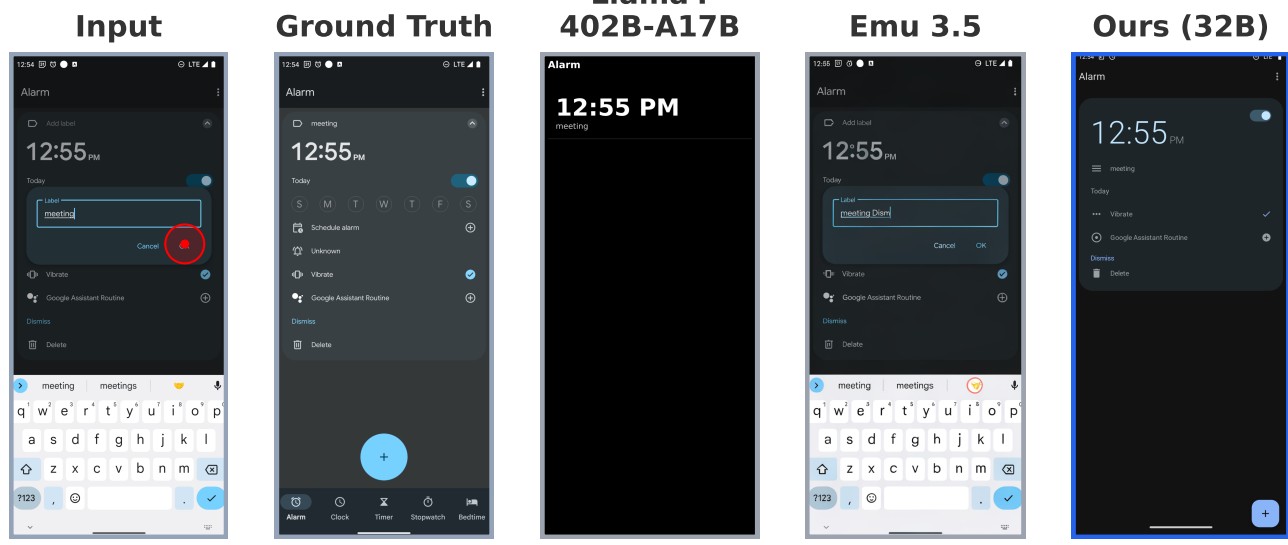

*Figure 16.* **Additional qualitative example 4.** The red marker on the input is for visualization only and was not provided to the model. Action: `click` at normalized coordinates (802, 394).

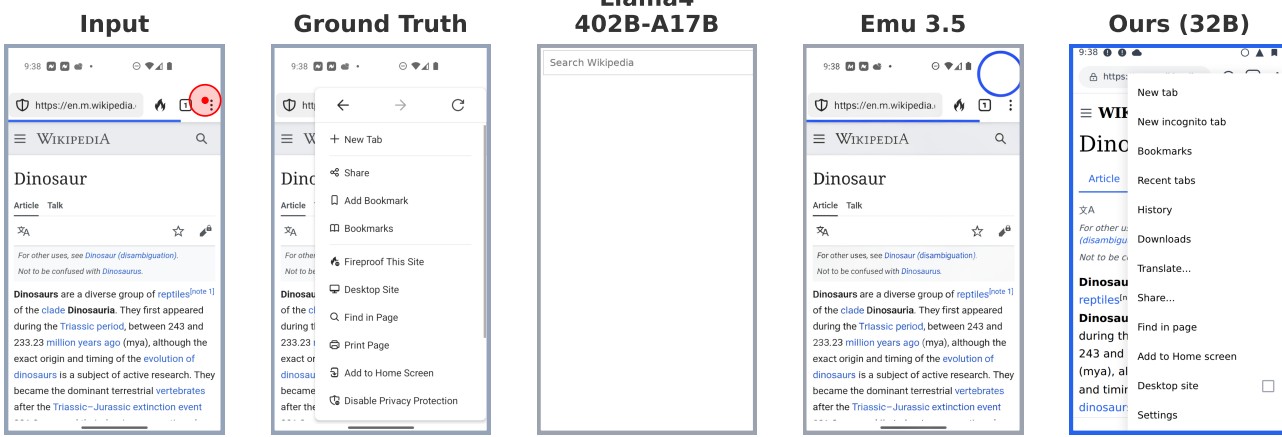

Figure 17. **Additional qualitative example 5.** The red marker on the input is for visualization only and was not provided to the model. Action: `click` at normalized coordinates (913, 143).

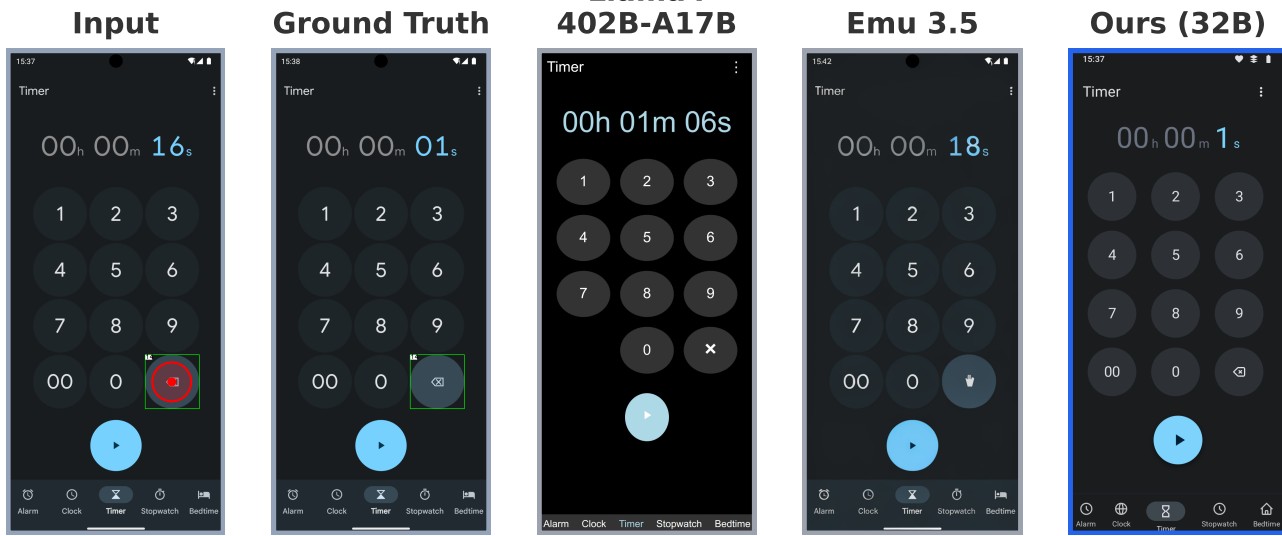

Figure 18. **Additional qualitative example 6 (Android World).** The red marker on the input is for visualization only and was not provided to the model. Action: `click` at normalized coordinates (756, 685).

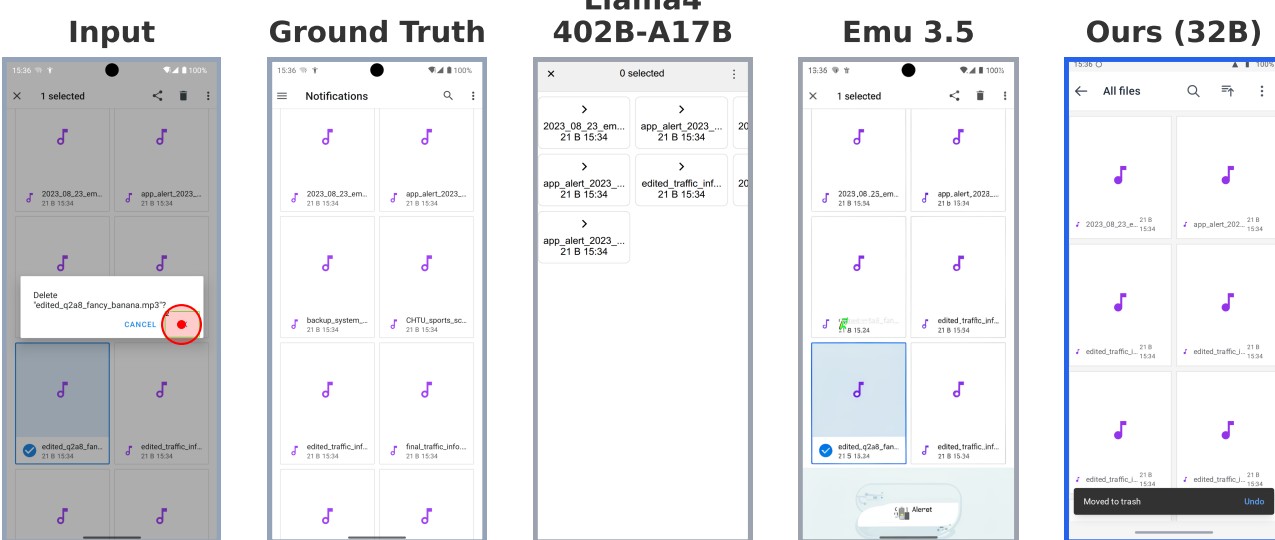

*Figure 19.* **Additional qualitative example 7 (Android World).** The red marker on the input is for visualization only and was not provided to the model. Action: `click` at normalized coordinates (819, 549).

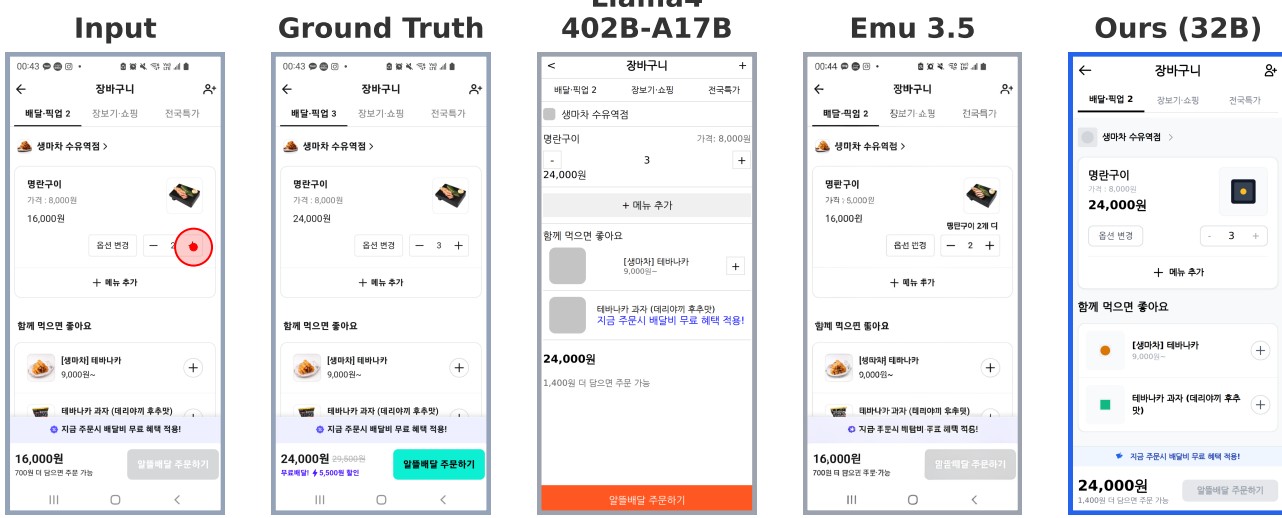

*Figure 20.* **Additional qualitative example 8.** The red marker on the input is for visualization only and was not provided to the model. Action: `TAP` at normalized coordinates (857, 421).

---

**Algorithm 3:** M3A + Value Function (No World Model)

---

**Input** : Test samples $\mathcal{D}^{\text{test}} = \{(S_t, A_t^{\text{GT}}, G, H_t)\}$ where $S_t$ is the current screenshot, $A_t^{\text{GT}}$ is the ground truth action, $G$ is the goal, and $H_t$ is the action history; policy model $\pi$; value function $V$; number of candidates $K$; prompts $P^{\text{alt}}, P^{\text{value-no-wm}}$

**Output** : Accuracy (rate of selecting ground truth)

correct $\leftarrow 0$;

**for** *each sample* $(S_t, A_t^{\text{GT}}, G, H_t) \in \mathcal{D}^{\text{test}}$ **do**

    /* (1) Build candidate set with GT as first candidate                         */

    $\mathcal{C} \leftarrow \{1 : A_t^{\text{GT}}\}$;

    /* (2) Generate $K - 1$ alternative actions                         */

    $\mathcal{A}^{\text{alt}} \leftarrow \pi(S_t, G, H_t, A_t^{\text{GT}}, P^{\text{alt}})$;

    **for** $i \leftarrow 2$ **to** $K$ **do**

        $\mathcal{C}[i] \leftarrow \mathcal{A}^{\text{alt}}[i - 1]$;

    /* (3) Score each candidate using value function (current state only)    */

    **for** *each* $i \in \{1, \ldots, K\}$ *in parallel* **do**

        $(v_i, c_i) \leftarrow V(S_t, \mathcal{C}[i], G, H_t, P^{\text{value-no-wm}})$;

    /* (4) Select valid action with highest confidence                    */

    $A_t^{\text{selected}} \leftarrow \arg\max_{i:v_i=\texttt{valid}} c_i$;

    **if** $A_t^{selected} = A_t^{GT}$ **then**

        correct $\leftarrow$ correct $+ 1$;

**return** $correct/|\mathcal{D}^{test}|$

---

---

**Algorithm 4:** M3A + World Model Evaluation

---

**Input** : Test samples $\mathcal{D}^{\text{test}} = \{(S_t, A_t^{\text{GT}}, G, H_t)\}$ where $S_t$ is the current screenshot, $A_t^{\text{GT}}$ is the ground truth action, $G$ is the goal, and $H_t$ is the action history; policy model $\pi$; world model $\mathcal{W}$; value function $V$; number of candidates $K$; prompts $P^{\text{alt}}, P^{\text{value-wm}}$

**Output** : Accuracy (rate of selecting ground truth)

correct $\leftarrow 0$;

**for** *each sample* $(S_t, A_t^{\text{GT}}, G, H_t) \in \mathcal{D}^{\text{test}}$ **do**

    /* (1) Build candidate set with GT as first candidate                         */

    $\mathcal{C} \leftarrow \{1 : A_t^{\text{GT}}\}$;

    /* (2) Generate $K - 1$ alternative actions                         */

    $\mathcal{A}^{\text{alt}} \leftarrow \pi(S_t, G, H_t, A_t^{\text{GT}}, P^{\text{alt}})$;

    **for** $i \leftarrow 2$ **to** $K$ **do**

        $\mathcal{C}[i] \leftarrow \mathcal{A}^{\text{alt}}[i - 1]$;

    /* (3) Predict next state for each candidate using world model             */

    **for** *each* $i \in \{1, \ldots, K\}$ *in parallel* **do**

        $S_{t+1}^{\text{code},(i)} \leftarrow \mathcal{W}(S_t, A_t, \mathcal{C}[i])$;

    /* (4) Score each (action, predicted next state) pair                 */

    **for** *each* $i \in \{1, \ldots, K\}$ *in parallel* **do**

        $(v_i, c_i) \leftarrow V(S_t, \mathcal{C}[i], S_{t+1}^{\text{code},(i)}, G, H_t, P^{\text{value-wm}})$;

    /* (5) Select valid action with highest confidence                    */

    $A_t^{\text{selected}} \leftarrow \arg\max_{i:v_i=\texttt{valid}} c_i$;

    **if** $A_t^{selected} = A_t^{GT}$ **then**

        correct $\leftarrow$ correct $+ 1$;

**return** $correct/|\mathcal{D}^{test}|$

---

| | Image-gen | | Code-gen | | | | | | | |
|---|---|---|---|---|---|---|---|---|---|---|
| | `Qwen-I-E` | `Emu3.5` | `Llama 4` | | | `Qwen3 VL` | | `GLM-4.6V` | `gWorld` | |
| **Model:** | | | | | | | | | | |
| **Parameter Size:** | **20B** | **34B** | **109B-A17B** | **402B-A17B** | **8B** | **32B** | **235B-A22B** | **106B** | **8B** | **32B** |
| **MWMBench-AitW** | | | | | | | | | | |
| IAcc.-`Gemini 3 Flash`(%,↑) | 10.4 | 12.8 | 33.6 | 36.3 | 15.2 | 35.4 | 32.4 | 50.1 | 68.2 | **69.6** |
| IAcc.-`GPT-5 Mini`(%,↑) | 17.8 | 31.0 | 54.2 | 54.0 | 22.8 | 52.9 | 39.8 | 68.0 | 76.0 | **81.2** |
| IAcc.-`Claude Haiku 4.5`(%,↑) | 18.0 | 26.4 | 55.1 | 51.4 | 26.4 | 52.0 | 36.1 | **64.6** | 62.2 | 64.2 |
| └ Render Fail (%,↓) | — | — | 4.4 | 9.4 | 33.8 | 11.6 | 40.0 | 2.4 | 0.8 | **0.6** |
| Similarity v1 (%,↑) | 70.8 | **79.9** | 71.4 | 72.9 | 61.5 | 71.1 | 75.3 | 76.6 | 77.6 | 78.6 |
| Similarity v2 (%,↑) | 49.4 | **57.4** | 44.4 | 44.9 | 38.2 | 46.8 | 50.5 | 52.7 | 54.9 | 55.9 |
| **MWMBench-GUIOdyssey** | | | | | | | | | | |
| IAcc.-`Gemini 3 Flash`(%,↑) | 7.4 | 15.5 | 37.5 | 40.0 | 22.2 | 45.0 | 50.0 | 65.6 | 79.0 | **87.2** |
| IAcc.-`GPT-5 Mini`(%,↑) | 17.9 | 30.5 | 63.2 | 67.5 | 31.8 | 60.7 | 61.2 | 76.8 | 85.0 | **87.4** |
| IAcc.-`Claude Haiku 4.5`(%,↑) | 13.8 | 31.5 | 58.6 | 60.0 | 30.6 | 50.2 | 53.0 | 62.2 | 67.6 | **69.8** |
| └ Render Fail (%,↓) | — | — | 1.2 | 7.8 | 51.4 | 16.0 | 27.2 | 3.8 | 1.2 | **0.8** |
| Similarity v1 (%,↑) | 74.2 | 80.7 | 77.6 | 79.1 | 59.4 | 75.0 | 81.6 | 83.7 | 83.9 | **84.7** |
| Similarity v2 (%,↑) | 53.4 | 56.9 | 47.0 | 48.9 | 37.2 | 50.3 | 57.7 | 61.3 | 62.7 | 62.6 |
| **MWMBench-AndroidControl** | | | | | | | | | | |
| IAcc.-`Gemini 3 Flash`(%,↑) | 4.2 | 17.4 | 33.4 | 45.6 | 26.0 | 45.4 | 51.4 | 74.6 | 81.2 | **86.8** |
| IAcc.-`GPT-5 Mini`(%,↑) | 19.9 | 32.8 | 56.0 | 65.8 | 35.2 | 61.0 | 55.9 | 80.8 | 84.6 | **85.4** |
| IAcc.-`Claude Haiku 4.5`(%,↑) | 11.0 | 32.9 | 62.7 | 64.3 | 32.1 | 53.3 | 48.3 | 67.1 | 69.3 | **76.5** |
| └ Render Fail (%,↓) | — | — | 1.0 | 8.6 | 42.8 | 13.4 | 34.2 | 1.4 | 2.6 | **0.8** |
| Similarity v1 (%,↑) | 76.4 | 81.2 | 75.6 | 78.6 | 63.1 | 75.3 | 80.3 | 82.8 | 83.2 | **84.8** |
| Similarity v2 (%,↑) | 51.2 | 55.9 | 47.2 | 47.6 | 43.6 | 52.9 | 57.3 | 60.0 | 62.4 | **63.5** |
| **MWMBench-AMEX** | | | | | | | | | | |
| IAcc.-`Gemini 3 Flash`(%,↑) | 3.2 | 12.0 | 33.2 | 51.2 | 26.6 | 51.6 | 48.7 | 68.9 | 84.3 | **90.6** |
| IAcc.-`GPT-5 Mini`(%,↑) | 18.2 | 27.4 | 54.9 | 64.7 | 35.6 | 58.8 | 54.0 | 72.8 | 86.8 | **87.2** |
| IAcc.-`Claude Haiku 4.5`(%,↑) | 11.2 | 25.8 | 58.8 | 59.1 | 39.0 | 60.4 | 50.8 | 66.8 | 76.8 | **80.4** |
| └ Render Fail (%,↓) | — | — | 0.6 | 12.6 | 31.6 | 3.8 | 30.0 | 1.2 | 0.8 | **0.4** |
| Similarity v1 (%,↑) | 77.0 | 84.2 | 81.8 | 82.7 | 70.8 | 82.0 | 83.6 | 84.7 | 84.8 | **85.6** |
| Similarity v2 (%,↑) | 51.8 | 58.9 | 52.0 | 53.4 | 47.6 | 57.9 | 59.8 | 61.6 | 63.8 | **65.1** |
| **MWMBench-AndroidWorld (out-of-distribution)** | | | | | | | | | | |
| IAcc.-`Gemini 3 Flash`(%,↑) | 8.0 | 25.8 | 32.4 | 41.3 | 26.8 | 44.8 | 47.7 | 70.7 | 72.8 | **79.9** |
| IAcc.-`GPT-5 Mini`(%,↑) | 15.9 | 35.9 | 63.9 | 63.6 | 34.3 | 63.1 | 59.6 | 81.9 | 83.1 | **85.9** |
| IAcc.-`Claude Haiku 4.5`(%,↑) | 17.6 | 25.5 | 56.7 | 58.0 | 31.2 | 52.2 | 46.1 | 69.7 | 69.1 | **73.9** |
| └ Render Fail (%,↓) | — | — | 2.9 | 14.4 | 42.3 | 13.1 | 30.0 | 1.9 | 2.3 | **0.4** |
| Similarity v1 (%,↑) | 81.1 | **87.3** | 76.2 | 76.5 | 60.9 | 74.2 | 77.7 | 84.1 | 81.2 | 83.1 |
| Similarity v2 (%,↑) | 54.7 | **61.0** | 47.2 | 47.0 | 38.8 | 48.1 | 52.6 | 60.3 | 57.2 | 60.1 |
| **MWMBench-Korean (out-of-distribution)** | | | | | | | | | | |
| IAcc.-`Gemini 3 Flash`(%,↑) | 8.7 | 10.1 | 32.5 | 45.7 | 24.6 | 45.1 | 59.2 | 52.1 | 65.5 | **77.2** |
| IAcc.-`GPT-5 Mini`(%,↑) | 23.0 | 36.6 | 59.0 | 72.9 | 34.9 | 61.8 | 74.3 | 67.7 | 75.4 | **83.6** |
| IAcc.-`Claude Haiku 4.5`(%,↑) | 15.4 | 33.8 | 53.7 | 61.2 | 30.7 | 50.5 | 59.0 | 52.3 | 61.4 | **66.3** |
| └ Render Fail (%,↓) | — | — | 1.8 | 2.2 | 38.8 | 8.1 | 15.4 | 4.4 | 0.8 | **0.6** |
| Similarity v1 (%,↑) | 83.0 | **83.5** | 71.8 | 73.7 | 60.9 | 74.2 | 78.0 | 74.7 | 76.1 | 76.3 |
| Similarity v2 (%,↑) | 58.9 | **58.9** | 42.7 | 43.4 | 40.0 | 51.4 | 56.6 | 52.7 | 56.0 | 56.0 |

*Table 12.* Granular results.

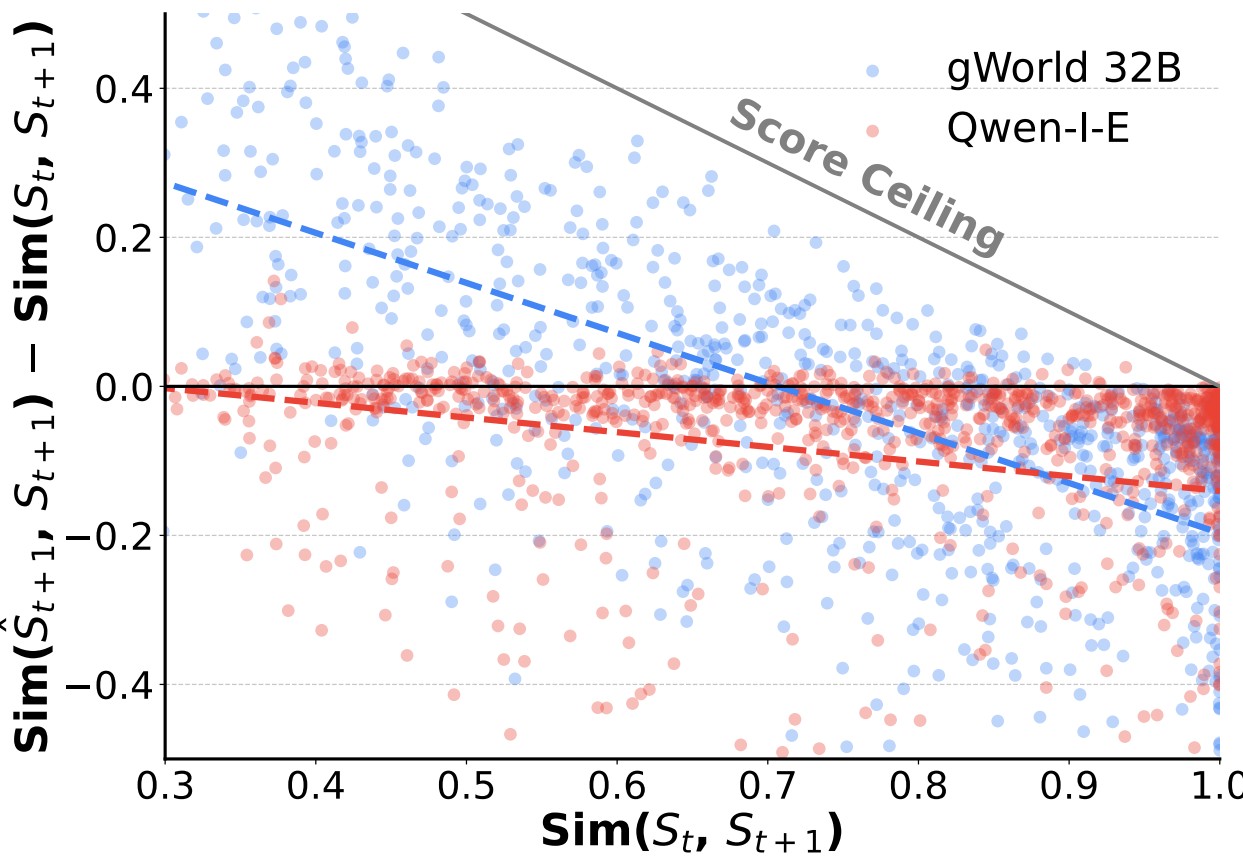

*Figure 21.* Same analysis as Figure 4 (bottom) with `Qwen-Image-Edit 20B` (`Qwen-I-E`). `Qwen-Image-Edit 20B` exhibits $\text{Sim}(\hat{S}_{t+1}, S_{t+1}) \approx \text{Sim}(S_t, S_{t+1})$, indicating that its outputs are nearly identical to the inputs ($S_t \approx \hat{S}_{t+1}$) regardless of the required action.

