# OpenReview forum: "Generative Visual Code Mobile World Models"
_ICML.cc/2026/Conference — ICML 2026 regular_

### Official Review · Reviewer_JGyg · 2026-03-12

**Soundness:** 3
**Presentation:** 2
**Significance:** 3
**Originality:** 4
**Overall Recommendation:** 4
**Confidence:** 4

**Summary:**

This paper proposes to design mobile world modelling (models that can simulate mobile GUI transitions) as a code generation task, which departs from pre-existing pixel-based or text-based approaches. The approach fine-tunes VLM for the task by building  a dataset that 1) leverages mobile agent training datasets, 2) translating GUI images into code using a state of the art pre-trained model, 3) generating reasoning text annotations. The approach is evaluated on a benchmark that assembles 6 datasets, including two out of distribution domains. Experiments show that the approach achieve strong results on the proposed benchmark. Authors also provide several detailed ablations, including a scale study.

**Compliance With Llm Reviewing Policy:**

Affirmed.

**Final Justification:**

The core contribution of this work is of great value, and I recommend accepting this manuscript. Authors have made substantial efforts to address my concerns and those of other reviewers, and I am now more confident of the work's value. I strongly encourage the authors to update their manuscript to incorporate the presentation improvement and additional experiments shared in the rebuttals.

**Key Questions For Authors:**

As mentioned in the weakness section, was there any quality control work done to evaluate the constructed dataset?

Could you clarify the connections between the proposed benchmark and datasets/benchmarks used in prior work? Is there overlap?

Could comparisons with VIMO be achieved by running experiments on a matching benchmark?

**Limitations:**

Authors provide a dedicated impact statement section, and have a detailed discussion of limitations in the conclusion. However, the latter is moved to the appendix, and should be part of the main paper.

**Strengths And Weaknesses:**

Strengths

Overall, I really like the idea of modelling this problem as a code generation task. It is an original and accurate solution to achieve the best of both worlds in terms of visual and textual accuracy. The methodology is simple, but doesn’t comprise any critical flaws or major concerns, and appears to work well.

The analysis is thorough, comparing across methods over 6 different domains. The impact of scale is also thoroughly studied, and a comprehensive set of ablations are provided. The impact of the proposed method on GUI agents capabilities is also studied, and the computational benefits to this formulation are verified.

The fact that this work plans to be open-source and encourages open research is appreciated as well.

Weaknesses

The presentation of the paper could be drastically improved. The introduction does not set the problem clearly enough: e.g. what does a mobile world model do? How does it help agents? What are GUI agents, what do they do?
The available space for the paper could be manage better, with e.g. figure 2 being very large, while critical components are in the appendix: conclusion and limitation, dataset details and evaluation.

The related works is extremely short and needs fleshing out to discuss the pros and cons of different approaches more in detail. While not essential as it is contemporary work, it would also be nice to relate to Code2World (Zheng et al, 2026) and highlight how the methods are related/differ.

The approach heavily relies on VLMs throughout: multiple times for dataset construction and for model evaluation, and seems to use the same models to build the dataset and evaluate results. This could risk introducing biases towards a model aligned with the prediction style of a specific VLM.

Regarding the dataset, authors do not mention any quality checks or data filtering to verify that the quality is good enough. Ensuring models are trained on good quality data is important, especially when their construction is heavily reliant on generated content.
While providing a new evaluation more thorough benchmark is a great contribution, it would be be good to provide results on at least one standard benchmark to verify 1) no biases towards the proposed setup/training data, 2) putting the work in context more clearly with prior works. This could also be a way to highlight what is missing from prior benchmarks (e.g. a method strong on one benchmark, but weak when looked at from another angle).

The ablation provided in Table 3 is very valuable, however, I think it would be more valuable as a model baseline, with a more thorough evaluation. This is particularly relevant as scores are relatively close. This would not necessarily affect the core contribution of modelling this problem as a code generation task, as the dataset construction enables to leverage less powerful open-source coding models such as QWEN.

Overall, I like the key contribution of the work and think it could be valuable to the community. I would be happy to increase my score if authors can address presentation issues and some limitations discussed above (e.g. Gemini coding model as a baseline, data quality control).

---

> ### Author Rebuttal · Authors · 2026-03-31
>
> Thank you for taking your time reviewing our paper. We will address your concerns and questions below. Your contribution has helped us improve the quality of our paper.
>
> ### **[W1] Improvements in presentation**
>
> We will use the extra camera-ready page to (i) move key components (conclusion, limitations, dataset details) into the main text, and (ii) clarify core concepts (GUI agents, world models) in the introduction.
>
> ### **[W2] Concurrent work and comprehensive related works**
>
> **[1]** Code2World, arXiv'd 13 days post ICML deadline, shares the use of renderable code. Key differences:
>
> **(i) Scalable supervision.** We focus on scalable data-centric supervision, demonstrating scaling laws ($R^2=0.95$) up to 260K samples with a pathway to 3.7M publically available transitions (**Tab. 8**). Code2World includes render-aware RL.
>
> **(ii) Decomposing reasoning and state generation; and look-ahead reasoning.** We uniquely decompose world modeling into reasoning + code generation, and introduce look-ahead reasoning traces grounded in the true next state.
>
> **(iii) Comprehensive benchmarking.** Our benchmark is the first to preserve the native action space; and our evaluation is a super-set of Code2World:
>
> |Benchmark|Modality|Action Space|In-Distribution|OOD|
> |:---|:---|:---|:---|:---|
> |MobileWorldBench|Text|Converted to Text|2× (AitW, AC)|-|
> |VIMO’s Benchmark|Visual|Converted to Text|2× (AitW, AC)|-|
> |Code2World|Visual|(Not explicitly specified)|1× (AC)|1× (GUIO)|
> |MWMBENCH (ours)|Visual|Original Coordinates + Text|4× (AitW, GUIO, AC, AMEX)|2× (AW, KA)|
>
> **[2]** We cover prior work across the following categories. Please let us know if there are any others we missed.
>
> |Topic|Line|No. of Literature|
> |:---|:---|:---|
> |Mobile GUI Agents|45-49|7×|
> |Text-based Mobile GUI World Models|33-38|7×|
> |Visual Mobile GUI World Model|40-96|1×|
> |Code-based World Models|131-137|4×|
> |Image to Web Code|139-145|6×|
> |Mobile UI Simulator|147-155|1×|
> |Mobile GUI World Model Benchmarking|195-266|2×|
>
> ### **[W3] Potential VLM-as-a-Judge bias has been carefully controlled and empirically validated**
>
> We mitigate VLM bias using three distinct frontier VLM families (**line 285–290**), with consistent results:
>
> * High inter-judge agreement (Spearman, Kendall; **Fig. 11**)
> * Invariant rankings across judges (`gWorld 32B/8B` always 1st/2nd; **Fig. 12**, **Tab. 9**)
>
> ### **[W4, Q2] Benchmarks are grounded in established literature and include two out-of-distribution evaluations**
>
> Our evaluation is grounded in standard literature. 5 of 6  benchmarks are directly repurposed from or built on established benchmarks.
>
> Moreover, our evaluation includes two out-of-distribution (OOD) benchmarks. Our method's strong, consistent performance across both in-distribution and OOD settings verifies that it learns generalized representations rather than exploiting artifacts of the training data.
>
> |Underlying Source|Venue|Citations|Distribution|
> |:---|:---|:---|:---|
> |Android in the Wild (AitW)|ICML 2023|366|In-Distribution|
> |GUIOdyssey (GUIO)|ICCV 2025|120|In-Distribution|
> |AndroidControl (AC)|NeurIPS 2024|118|In-Distribution|
> |Android Multi-annotation Expo (AMEX)|ACL Findings 2025|97|In-Distribution|
> |AndroidWorld (AW)|ICLR 2025|256|Out-of-Distribution|
> |KApps (KA)|Newly introduced|--|Out-of-Distribution|
>
> Finally, as shown in **Tab. 1**, we situate our work within prior literature by demonstrating how our broader evaluation exposes failure modes that methods over-optimized for older, narrower benchmarks miss.
>
> ### **[W5] Ablations validate components; baselines are evaluated separately**
>
> We follow the *standard* academic practice of using established method and models as baselines in the main results, while reserving ablations to analyze individual components. Our ablation study isolates the contributions of Step 1 and Step 2 in **Tab. 3**, and Step 3 in **Fig. 6** relative to a naïve alternative.
>
> For a complete decomposition of our method, refer to **[Q1, Q3] Baselines appropriately answer scientific hypotheses by decomposing performance gains** in Response to `D2FN`
>
> ### **[Q1] Dataset quality**
>
> We did not apply explicit filtering, as preliminary analysis showed negligible error rates. See **[Q5] Quantifying the quality of synthetic reasoning traces** in Response to `D2FN`.
>
> ### **[Q3] Performance and speed comparison with VIMO**
>
> As noted in **Sec. 5**, `gWorld` achieves 81–81.9% vs. VIMO’s 74%, with 123×–291× end-to-end speedups. See **[Q5] Our method's end-to-end inference is up to 291x faster than VIMO** in Response to `w2bh`.
>
> ### **[Additional Result During Rebuttal]**
>
> Please refer to **[W5] Human evaluation study** in Response to `qrZ1`.

---

> > ### Author Rebuttal · Reviewer_JGyg · 2026-04-03
> >
> > I thank the authors for their detailed response, that has addressed most of my concerns. As mentioned in my original review, I find the core contribution is of enough significance to support acceptance. I strongly encourage the authors to incorporate the additional experiments and details provided in the rebuttals, and to make efforts to improve presentation in the final version.

---

> > > ### Author Response · Authors · 2026-04-06
> > >
> > > Thank you for your constructive comments and for confirming that our rebuttal has adequately addressed your concerns.
> > >
> > > We will ensure all new results and discussion points are incorporated into the revised manuscript to further improve the paper's clarity.
> > >
> > > Best,
> > > Authors

---

### Official Review · Reviewer_qrZ1 · 2026-03-18

**Soundness:** 3
**Presentation:** 3
**Significance:** 2
**Originality:** 3
**Overall Recommendation:** 3
**Confidence:** 3

**Summary:**

This work introduces a new formulation for modeling mobile GUI dynamics by predicting the next interface state as executable web code rather than pixel-level images. The proposed model, gWorld (8B/32B), is a vision-language system trained on a synthetic dataset derived from mobile interaction trajectories. In this pipeline, future UI states are transformed into code representations and enriched with reasoning traces. Experimental results demonstrate strong performance across several benchmarks, including out-of-distribution settings, with clear improvements over existing image-based and VLM-based approaches in terms of IAcc. The authors also show that higher-quality world models can improve downstream GUI agent performance through simple planning strategies.

**Compliance With Llm Reviewing Policy:**

Affirmed.

**Key Questions For Authors:**

Please see the above weakness.

**Limitations:**

Yes

**Strengths And Weaknesses:**

Strengths:
1. Representing GUI transitions as executable code instead of pixels is both intuitive and compelling. This approach addresses known limitations of image generation, such as poor text rendering and structural inconsistencies, and reflects a meaningful shift in representation design.
2. The method achieves consistent gains across multiple datasets, including OOD scenarios, and outperforms significantly larger baselines on IAcc. The experimental evaluation is thorough and convincing.
3. The paper presents a well-structured three-stage data construction pipeline. Ablation studies effectively demonstrate the contributions of individual components, such as reasoning traces and code relabeling.

Weaknesses:
1. A key concern is that the training data is generated using a powerful proprietary model (e.g., Gemini 3 Flash), which is responsible for: converting images into code representations, producing reasoning traces with access to the ground-truth next state. This raises the possibility that much of the performance gain comes from implicit distillation rather than the proposed modeling approach. The paper does not clearly separate: 1) the benefits of using code as a representation; 2) the benefits of high-quality supervision from a strong teacher model. Without stronger controls (e.g., weaker teachers or human-annotated data), it is difficult to attribute the improvements correctly.

2. Look-ahead reasoning introduces information leakage
The reasoning traces are generated with access to the true next state, which is not available at inference time. Although framed as supervision, this effectively injects oracle information into training targets. This setup may further amplify the distillation effect and obscure the true source of performance gains.

3. The policy improvement experiments rely on the learned world model to simulate future states. However, if the world model has already absorbed knowledge from a strong teacher, the observed gains may not purely reflect better transition modeling. Instead, they may result from: 1) transferred capabilities from the teacher model; 2) richer semantic information embedded during data generation. Although a correlation is shown between world model quality and policy performance, it remains unclear whether this is due to improved modeling or inherited knowledge. This weakens the causal claims in Section 4.6.

4. While the code-based representation is promising, the paper does not sufficiently explore other structured formats, such as UI trees or layout graphs. It is therefore unclear whether web code is uniquely advantageous or simply one of several viable options.

5. The main metric, IAcc, depends on VLM-based judges. Even with multiple evaluators, this introduces potential bias and limits interpretability. Incorporating human evaluation or task-level success metrics would strengthen the claims.

---

> ### Author Rebuttal · Authors · 2026-03-31
>
> Thank you for taking your time reviewing our paper. We will address your concerns and questions below. Your contribution has helped us improve the quality of our paper.
>
> ### **[W1, W3] Disentangling representation + synthesis pipeline vs. distillation effects**
>
> Our gains are not explained by implicit distillation, but by a novel representation + scalable data generation pipeline, each independently validated.
>
> **[1] Decomposition of representation shift vs. synthesis pipeline.** Please refer to **[Q1, Q3] Baselines appropriately answer scientific hypotheses by decomposing performance gains** in Response to `D2FN`.
>
> **[2] Explicit controls against implicit distillation.** We granularly isolate `gWorld`'s gains over pure distillation baselines. For state generation, `gWorld`'s Steps 1-2 outperform distillation, achieving perfect (100%) code renderability and a +1.9~5.4% data quality gain (**Tab. 3**). Similarly, for reasoning trace generation, `gWorld`'s Step 3 outperforms distillation, yielding consistent gains across all 5 benchmarks (**Fig. 6**).
>
> **[3]** Using strong models as annotators is a standard paradigm (see **[W2] Leveraging frontier models is an established paradigm** in Response to `D2FN`).
>
> ### **[W2, W3] No test-time leakage; this follows the well-established “learning with privileged information” paradigm**
>
> **[1]** The future state $S_{t+1}$ is only used during *training* to curate supervision for $R_t$, and is never available to the model during evaluation. Rather, our approach follows a well-established paradigm *Learning Using Privileged Information (LUPI)*:
>
> |Paper|LUPI|Venue|
> |:---|:---|:---|
> |A New Learning Paradigm: Learning Using Privileged Information|Establishes the theoretical foundation showing that access to additional training-only signals improves generalization.|Neural Networks 2009|
> |Bidirectional RNNs (e.g., BiLSTM)|During training, models condition on both past and *future* context, while at inference tasks (e.g., causal generation) only past context is available. This is a canonical example of leveraging future information for better representations.|NAACL 2018|
> |STaR: Bootstrapping Reasoning With Reasoning|Uses ground-truth final answers (future information) in the *prompt* during training to improve reasoning via rejection sampling.|NeurIPS 2022|
> |Provable Partially Observable Reinforcement Learning with Privileged Information|Provides guarantees in sequential decision-making where training uses information unavailable at test time.|NeurIPS 2024|
>
> **[2]** Consequently, as policy evaluation is also on held-out test sets, gains cannot be explained by memorized distilled knowledge.
>
> ### **[W4] Choice of visual representation vs. other structured formats**
>
> We clarify why visual code is uniquely advantageous in our setting and why other structure representations are *orthogonal* directions.
>
> **[1] Information preservation.** Our approach prioritizes *visual fidelity*. Unlike UI trees, which are inherently abstracted and discard fine-grained attributes, visual code-based representations (HTML/CSS) preserve both structural and visual information. As discussed in **Sec. 1** (**line 40–48**), prior text-based abstractions lose spatial layout, iconography, typography, and color, which are essential for accurate interaction modeling in real-world environments.
>
> **[2] Alignment with VLM-based policies.** Modern VLMs operate directly over visual inputs and consistently outperform language-only approaches on GUI tasks. By retaining a representation that faithfully renders back into pixel space, our method enables tighter coupling between the world model and VLM-based policies.
>
> **[3] Structured representations are orthogonal and complementary.** They can be incorporated on top of our visual world model to further enhance structured reasoning or planning.
>
> ### **[W5] Human evaluation study**
>
> Following `w2bh`'s suggestion, we conduct a human evaluation on 100 equal-weighted random samples across all 6 benchmarks, with 12 annotators, yielding 300 annotations in total and 3 independent annotation per sample. Given the current UI, action, the ground-truth next state, and 4 randomized candidate outputs, annotators ranked all 4 outputs from best to worst. We report average rank. Human judgments support our conclusions: `gWorld 32B` ranks first overall (1.68 avg. rank, 77.5% pairwise win rate), followed by `gWorld 8B` (2.16, 61.4%). Human and VLM-as-a-Judge agreement is high, with Spearman ρ = 0.806 and Kendall τ = 0.600.
>
> |Model|Avg. Rank ↓|IAcc (%) ↑|
> |:---|---:|---:|
> |gWorld 32B|1.68|79.6|
> |gWorld 8B|2.16|74.9|
> |Qwen3 VL 235B|2.19|51.5|
> |GLM-4.6V 106B|2.21|67.4|
> |Qwen3 VL 32B|2.52|52.5|
> |Llama 4 402B|2.58|55.7|
> |Qwen-I-E 20B|2.73|13.4|
> |Emu3.5 34B|2.77|25.8|
> |Llama 4 109B|2.98|50.0|
> |Qwen3 VL 8B|3.19|29.2|
>
> ### **[Additional Result During Rebuttal]**
>
> Please consider additional results in **[Q5] Our method's end-to-end inference is up to 291x faster than VIMO** in Response to `w2bh`.

---

> > ### Author Rebuttal · Reviewer_qrZ1 · 2026-04-05
> >
> > Thank the authors for the detailed response, which has addressed most of my concerns. I decide to raise my rating.

---

> > > ### Author Response · Authors · 2026-04-06
> > >
> > > Thank you for your constructive comments and for confirming that our rebuttal has adequately addressed your concerns.
> > >
> > > We will ensure all new results and discussion points are incorporated into the revised manuscript to further improve the paper's clarity.
> > >
> > > Best,
> > > Authors

---

### Official Review · Reviewer_w2bh · 2026-03-19

**Soundness:** 3
**Presentation:** 3
**Significance:** 2
**Originality:** 3
**Overall Recommendation:** 4
**Confidence:** 3

**Summary:**

This paper proposes gWorld, a mobile GUI world model that predicts the next screen state as renderable web code (HTML) rather than generating pixels directly or outputting text descriptions. The key insight is that VLMs' pre-training on web code enables high-fidelity visual generation of GUI states while preserving text rendering accuracy — a known weakness of image-generation approaches. The authors build a data generation pipeline that repurposes offline policy trajectories into (state, action) → (reasoning, code) training pairs, with a cross-modal re-labeling step (pixel → HTML) and look-ahead reasoning traces. gWorld (8B, 32B), fine-tuned from Qwen3 VL, is evaluated on MWMBENCH — a new suite of 4 in-distribution and 2 out-of-distribution benchmarks — and achieves state-of-the-art instruction accuracy while being significantly smaller than competing models (e.g., outperforming Llama 4 402B and GLM-4.6V 106B).

**Compliance With Llm Reviewing Policy:**

Affirmed.

**Final Justification:**

I appreciate the authors’ detailed rebuttal and the additional experiments. After reading the rebuttal and follow-up responses, I am persuaded that the paper is stronger than I initially assessed, so I am **raising my score from 3 to 4**.

The rebuttal **materially improved** the paper for me, especially by adding **human evaluation** and more concrete **practical analysis**.

But, I still have **important reservations**. I continue to view the work primarily as a strong **representation and data-engineering contribution** rather than a fundamentally new learning method. The dependence on a **strong proprietary teacher** also remains a real weakness: while the rebuttal clarifies the pipeline and argues that this is a standard paradigm, it still does not show how sensitive the method is to **weaker or open-weight alternatives**, so the source of the gains is not fully disentangled. I also remain cautious about the **downstream claims**. The original policy experiment used an **oracle-style setting**, and although the follow-up non-oracle results are helpful, the gains there are much smaller, which suggests that the practical impact on realistic deployment may be more limited than the headline result initially suggests. For these reasons, while I am now on the accept side, I remain **only weakly positive**.

**Key Questions For Authors:**

1. Can you provide a human study validating IAcc. on your benchmarks? The cited correlation from VIMO is on a different benchmark with different evaluation setup. Even a small-scale human annotation (e.g., 100 samples) comparing human judgment with the 3-judge IAcc. would significantly strengthen the claims.

2. How does the quality of the frontier model used for data generation affect downstream gWorld performance? Have you tried using different frontier models (e.g., open-weight alternatives) for the image-to-code conversion step? This is crucial for reproducibility.

3. Can you show policy evaluation results in a non-oracle setting (e.g., K candidates all generated by the policy, without guaranteed ground-truth inclusion)? The oracle K=3 setting may not reflect real deployment conditions.

4. What fraction of transitions in your benchmarks involve photo-realistic content (camera, video, complex images), and what is gWorld's IAcc. specifically on those transitions vs. text/UI-heavy transitions?

5. The paper claims VIMO's multi-stage pipeline has "significant computational overhead and latency," but does not report gWorld's inference latency. What is the end-to-end latency for gWorld (VLM inference + code rendering) vs. baselines?

**Limitations:**

Partially addressed. The authors mention photo-realistic content limitations and the single-frame Markov assumption in the appendix (page 24), but these are brief. Missing discussion of: (a) dependency on proprietary frontier models for data generation, (b) the evaluation metric's reliance on VLM judges without human validation on their benchmark, (c) potential failure modes when mobile UIs use non-standard widgets or animations.

**Strengths And Weaknesses:**

**Strengths**

- S1: The code-as-visual-representation paradigm is a genuinely creative reframing. Using HTML generation to bypass the pixel-generation bottleneck for structured GUIs is well-motivated and non-obvious. It elegantly sidesteps the text rendering problem that plagues diffusion-based approaches.

- S2: Comprehensive evaluation across 6 benchmarks (4 ID + 2 OOD), 10 baselines, and 3 independent VLM-as-a-Judge models (GPT-5 Mini, Claude 4.5 Haiku, Gemini 3 Flash) with high inter-judge agreement. The multi-judge setup is rigorous and mitigates model-family bias.

- S3: Clear paper structure, well-designed figures (especially Fig. 2 showing qualitative comparisons), and thorough appendix with full prompts and algorithms.

**Weaknesses**

- W1: The primary evaluation metric (IAcc.) relies entirely on VLM-as-a-Judge, which introduces potential systematic biases. Although 3 judges are used and inter-judge agreement is reported, the correlation with human judgment is only cited from Luo et al. (2025) for a different setting (VIMO's benchmark). No human evaluation is conducted on MWMBENCH itself to validate the metric. Given this is a new benchmark, this is a notable gap.

- W2: The data generation pipeline depends heavily on a "frontier model π*" for both image-to-code conversion and reasoning trace generation. The specific model used is not disclosed (anonymity reasons), but this means: (a) training data quality is upper-bounded by a proprietary model's capabilities, (b) the approach's reproducibility is limited, and (c) it's unclear how sensitive results are to the choice of π*. No ablation on different π* models is provided.

- W3: The policy evaluation (Tab. 4) uses an oracle setup where K=3 candidates include the ground-truth action. This significantly simplifies the selection task and may overstate real-world gains. With K=3, random baseline accuracy is 33%. The +22% gain over Qwen3 VL 8B is measured in this oracle setting — real deployment gains could differ substantially.

- W4: The approach has an inherent limitation acknowledged but under-discussed: web code cannot faithfully represent photo-realistic content (videos, complex images, camera views). Many real mobile GUI states involve such content. The paper does not quantify what fraction of real-world transitions this affects or how badly it degrades in those cases.

- W5: The practical value proposition of a GUI world model over a lightweight simulator is not convincingly established. Mobile apps can be deployed in Docker/emulator sandboxes at low cost and high fidelity — providing ground-truth next states with no approximation error. The paper's data pipeline itself requires a strong frontier model (π*) for image-to-code conversion, which is expensive. Yet no latency or cost comparison is made against a simulator-based alternative. More critically, the downstream usage (§4.6) is limited to a single-step oracle action selection (K=3, breadth-only rollout with very short horizon), rather than using the world model for multi-step trajectory generation, data augmentation, or actual agent training via imagination (e.g., Dyna-style). This significantly narrows the demonstrated significance — the world model serves as a marginal value function augmentation rather than enabling fundamentally new training or planning capabilities.

---

> ### Author Rebuttal · Authors · 2026-03-31
>
> Thank you for taking your time reviewing our paper. We will address your concerns and questions below. Your contribution has helped us improve the quality of our paper.
>
> ### **[W1, Q1] Human evaluation validates IAcc. and confirms gWorld ranking**
>
> Please refer to **[W5] Human evaluation study** in Response to `qrZ1`.
>
> ### **[W2] Frontier model disclosure, quality, and reproducibility**
>
> **[1] Disclosure & paradigm:** The frontier model (Gemini 3 Flash) is disclosed in **Sec. 2.5**. Using strong models as annotators is a standard paradigm (see **[W2] Leveraging frontier models is an established paradigm** in Response to `D2FN`).
>
> **[2] Reproducibility:** Our framework is model-agnostic. Using a cost-efficient model provides a conservative lower bound; stronger models would likely improve results. Full reproducibility is ensured via prompts and pipeline details in the Appendix.
>
> ### **[W3, Q3] Measuring potential policy gains**
>
> Please refer to **[Q4] Measuring potential policy gains** in Response to `D2FN`.
>
> ### **[W4, Q4] Impact of photo-realism and dataset composition**
>
> Semantic and structural fidelity drive task success far more than the photo-realism of embedded media.
>
> When photo-realism is present, gWorld’s SVG approximations successfully retain sufficient shape and color profiles for interaction (see **Fig. 20**). This theoretical trade-off is quantitatively validated in **Tab. 2**, where gWorld achieves the highest embedding similarity (a standard metric for perceptual fidelity; Zhang et al., CVPR 2018; Ruiz et al. CVPR 2023), proving that our structural accuracy gains outweigh localized losses in photo-realism.
>
> To answer [Q4], we classified all in-distribution test samples using a frontier VLM (Gemini 3 Flash). Photo-realistic content constitutes a minority of the benchmark (17.4%), while the majority (81.5%) are UI/text-heavy transitions. While gWorld exhibits a slight performance (IAcc.) dip on photo-realistic transitions, it maintains a massive lead over all baselines in both categories. This confirms that gWorld's advantage is robust and not an artifact of a dataset skewed toward text-heavy UIs.
>
> |Model|Photo-realistic (17.4%)|Others (81.5%)|Δ|
> |:---|:---|:---|:---|
> |gWorld 8B|75.26%|75.92%|-0.66%|
> |gWorld 32B|74.98%|79.68%|-4.70%|
> |Qwen3 VL 8B|25.02%|29.34%|-4.32%|
> |Qwen3 VL 32B|49.19%|52.92%|-3.73%|
> |Qwen3 VL 235B|50.05%|48.03%|+2.02%|
> |Llama 4 402B|54.35%|54.80%|-0.45%|
> |GLM-4.6V 106B|67.91%|67.83%|+0.08%|
> |Emu3.5 34B|22.92%|24.90%|-1.98%|
> |Qwen-Image-Edit 20B|14.04%|12.56%|+1.48%|
>
> ### **[W5] Advantages of a generative world model vs. real device**
>
> While real devices and emulators provide ground-truth states, they suffer from fundamental bottlenecks in tree search, scaling, and generalization.
>
> **[1] Unlocking efficient tree search via state branching:** Real devices face limitations during search due to irreversible state transitions. Exploring $K$ branches from $S_t$ on a device requires reversing actions. Often, there is no trivial inverse; e.g., the "back" button does not undo adding an item to a cart; reverting requires multiple steps or slowly replaying the entire history from $S_0$. World models enable instant, arbitrary state branching without environment resets.
>
> **[2] Scaling & Latency:** Docker-based emulators are CPU/RAM bottlenecked and require cumbersome infrastructure. Resetting to a new initial state $S_0$ incurs a $>2$s overhead, and Android Virtual Device (AndroidWorld; ICLR 2025) require a 1s buffer for transitions to finish. In contrast, gWorld leverages highly parallelizable GPU batched inference. Our end-to-end latency per transition is highly competitive: 0.55s for 8B and 1.3s for 32B on 4 GPUs. This makes massive, batched RL rollouts easily scalable.
>
> **[3] Generative generalization:** Simulators cannot extrapolate beyond programmed boundaries. Generative models synthesize novel transitions (see OOD performance in **Tab. 2**, **Fig. 5**). This probabilistic generation can make policy training robust to diverse visual variations and unseen apps.
>
> **[4] Downstream scope:** The single-step search ($K=3$) is a foundational proof-of-concept following VIMO (ICLR 2026).
>
> ### **[Q5] Our method's end-to-end inference is up to 291x faster than VIMO**
>
> Reported in Sec. 5 (line 401-415), on 4 GPUs, gWorld's end-to-end latency per state is 1.3s (32B) and 0.55s (8B), combining a single VLM pass (1.0/0.25s) and lightweight rendering (0.3s).
>
> Conversely, VIMO reports 160s. VIMO is bottlenecked by its sequential multi-stage pipeline (OCR, masking, diffusion, etc.) rather than raw compute. Even assuming instant local model execution (saving 38s; reported in their paper) via improved GPUs, VIMO's remaining 122s structural overhead is 93× slower than gWorld 32B.
>
> |Model|End-to-End Latency (s)|Speed-up vs. VIMO (×)|
> |:---|:---|:---|
> |gWorld 8B|0.55|291×|
> |gWorld 32B|1.30|123×|
> |VIMO|160|-|
> |VIMO (idealized, -38s)|122|1.31×|

---

> > ### Author Rebuttal · Reviewer_w2bh · 2026-04-03
> >
> > The rebuttal is helpful and addresses several practical concerns, but I still have two follow-up questions. First, the downstream policy experiment remains a single-step oracle evaluation: the paper measures “potential” gains with a K=3 candidate set that includes the ground-truth action, and the rebuttal explicitly frames this as a proof-of-concept. I therefore still find it unclear how much of the reported gain would transfer to realistic non-oracle deployment or to broader training/planning use cases. Second, although the rebuttal now discloses the frontier model used for data generation, the sensitivity of the method to that teacher choice is still unclear because no weaker or open-weight alternative is evaluated.

---

> > > ### Author Response · Authors · 2026-04-05
> > >
> > > We appreciate the reviewer for the thoughtful follow-up questions.
> > >
> > > ### **[Q1] Additional world model enhanced policy results**
> > > Here are results without ground truth included in the $K$ action candidates. We will perform a larger scale experiment with more backbone models, and integrate results in the paper.
> > >
> > > **Table 1: gWorld enhanced policy improvement**
> > > | Policy  | w/o WM | WM | $\Delta$ |
> > > | :--- | :--- | :--- | :--- |
> > > | Gemini-2-flash  | 32.0% | 35.0% | **+3.0%** |
> > > | Gemini-2.5-flash  | 33.0% | 34.0% | **+1.0%** |
> > >
> > > As mentioned previously, while the world model provides gains, its efficacy is constrained by the base policy’s `pass@k`. To investigate this, we analyzed cases where the baseline policy failed and measured how often the correct action was present in the alternative candidates.
> > >
> > > | Category | Proportion |
> > > | :--- | :---: |
> > > | Correct answer in alternative candidates |12% |
> > > | Correct answer **absent** from all candidates| 88% |
> > >
> > > Our original oracle setting isolated this *candidate sampling* issue. This means that this orthogonal problem will improve as backbone policies improve over time. Training policies to generate more diverse and correct candidates is a *complementary* and *orthogonal* research challenge.
> > >
> > > ### **[Q2] Frontier model choice**
> > >
> > > We agree that frontier model sensitivity is an important consideration.
> > >
> > > **[1]** We emphasize that our framework is *model-agnostic* and relies on a frontier model for **scalable** Image-to-web-code **annotation**. Gemini 3 Flash (3 dollar/M tokens generated) is much cheaper than other alternatives like Gemini 3 Pro (12 dollar/M tokens generated) and Claude Opus 4.6 (25 dollar/M tokens generated). Meaning, performance will likely improve when using a better frontier model. Moreover, given the SoTA performance of open-weight models like GLM-4.5V on web code generation benchmarks (e.g., Design2Code), we expect our methodology to generalize across different high-performing VLM models, especially as the field continues to evolve.
> > >
> > > **[2]** Notably, this is an integral paradigm embraced by the research community (e.g., in ICML, ICLR).
> > >
> > > Leveraging an existing strong model for data curation and synthesis is standard practice for training state-of-the-art (SoTA) models. Examples include:
> > > 1. RLAIF (ICML 2024) leverages frontier LLMs to provide reward signals.
> > > 2. Qwen and DeepSeek curate training data via frontier model-based filtering (Qwen2.5, 2.5 VL, 3, DeepSeek-R1; 2025).
> > > 4. WizardCoder (ICLR 2024) and Magicoder (ICML 2024) leverage frontier models to synthesize, relabel, and evolve complex coding instructions.
> > >
> > > **[3]** Finally, frontier models have improved in performance and decreased in cost over time, ensuring our method is scalable and replicable (Stanford AI Index; 2025).

---

### Official Review · Reviewer_D2FN · 2026-03-21

**Soundness:** 3
**Presentation:** 3
**Significance:** 2
**Originality:** 2
**Overall Recommendation:** 4
**Confidence:** 1

**Summary:**

This paper proposes gWorld which predicts the next mobile GUI state as renderable web code instead of pixels.
The training pipeline repurposes offline policy trajectories and relabels next states into web code and also adds look ahead reasoning traces.
The paper also introduces MWMBENCH which covers four in distribution benchmarks and two out of distribution benchmarks in the native visual setting.
The reported results show that gWorld 8B and 32B outperform much larger open weight baselines on the main benchmark suite.

**Compliance With Llm Reviewing Policy:**

Affirmed.

**Key Questions For Authors:**

How much of the gain comes from the renderable code representation itself rather than the closed model data synthesis pipeline.

Would the main conclusions still hold under a larger human evaluation instead of mostly relying on VLM judges.

Why is there no stronger apples to apples baseline where competing models also receive similar task specific post training.

How much policy gain remains when the ground truth action is removed from the K equals 3 candidate set.

Can the authors quantify the error rate of the synthetic reasoning traces before they are used for training.

**Limitations:**

Yes

**Strengths And Weaknesses:**

S1: The code based world modeling idea is intuitive and it directly targets the text rendering and layout problems of image generation baselines.

S2: The paper does a decent job on ablations and scaling and these experiments make the engineering story more believable.

W1: The main contribution feels more like representation design plus data bootstrapping than a clearly new learning method.

W2: The whole training pipeline depends heavily on Gemini 3 Flash for code relabeling and reasoning synthesis.

W3: The main evaluation still leans on VLM judges so the headline metric is not fully grounded in direct human assessment.

---

> ### Author Rebuttal · Authors · 2026-03-31
>
> Thank you for taking your time reviewing our paper. We will address your concerns and questions below. Your contribution has helped us improve the quality of our paper.
>
> ### **[W1] Our principal contribution is significant**
>
> Valuable contributions to the community encompass a wide range of advancements and do not strictly require a new learning algorithm. Consider the Main Track Reviewer Instructions, which states:
> > "be open-minded about the potential strengths and **broad definitions** of significance and originality"
>
> Our key contribution is a unified paradigm that (i) enables executable visual state modeling by shifting from pixel to renderable code space, and (ii) a scalable automatic data synthesis pipeline.
>
> Other reviewers recognized our approach as significant and impactful:
> * **Novelty**
>     * "representation paradigm is a genuinely creative reframing" (`w2bh`)
>     * "non-obvious" (`w2bh`)
>     * "intuitive and compelling" (`qrZ1`)
>     * "meaningful shift in representation design" (`qrZ1`)
>     * "an original and accurate solution" (`JGyg`)
>     * "really like the idea of modelling this problem as a code generation task." (`JGyg`)
> * **Clear Motivation**
>     * "is well-motivated" (`w2bh`)
>     * "addresses known limitations of image generation" (`qrZ1`)
> * **Impact**
>     * "impact of the proposed method on GUI agents capabilities is also studied" (`JGyg`)
>     * "valuable to the community" (`JGyg`)
>
> ### **[W2] Leveraging frontier models is an established paradigm**
>
> While our method utilizes Gemini 3 Flash, this is an integral paradigm embraced by the research community (e.g., in ICML, ICLR).
>
> First, leveraging an existing strong model for data curation and synthesis is standard practice for training state-of-the-art (SoTA) models. Examples include:
> 1. RLAIF (ICML 2024) leverages frontier LLMs to provide reward signals.
> 2. Qwen and DeepSeek curate training data via frontier model-based filtering (Qwen2.5, 2.5 VL, 3, DeepSeek-R1; 2025).
> 4. WizardCoder (ICLR 2024) and Magicoder (ICML 2024) leverage frontier models to synthesize, relabel, and evolve complex coding instructions.
>
> Second, frontier models have improved in performance and decreased in cost over time, ensuring our method is scalable and replicable (Stanford AI Index; 2025).
>
> ### **[W3, Q2] Human evaluation validates IAcc. and confirms gWorld ranking**
>
> Please refer to **[W5] Human evaluation study** in Response to `qrZ1`.
>
> ### **[Q1, Q3] Baselines appropriately answer scientific hypotheses by decomposing performance gains**
>
> Our baseline setup was designed to isolate and validate our  hypotheses. The larger models are included as performance references.
>
> * **Hypothesis 1:** Visual-code-based representation is superior to image-generation representation, given current SoTA open-source models.
> * **Hypothesis 2:** Our automatic training data synthesis methodology effectively improves world modeling capabilities.
>
> Assuming that SoTA open-weight image-generation models and VLMs are a good approximation of baseline representation capabilities, we compare `Qwen-Image-Edit 20B` and `Emu3.5 34B` against a similar-sized VLM, `Qwen3 VL 32B`. Then, moving from `Qwen3 VL` to `gWorld` isolates gains achieved by our automatic training data generation.
>
> |Model|Average IAcc. (%)|Gains from Representation Shift (Δ)|Gains from Data Gen (Δ)|
> |:---|:---|:---|:---|
> |Qwen3 VL 32B|52.5|+26.7% (vs. Emu3.5),+39.1% (vs. Qwen-I-E)|-|
> |gWorld 8B|74.9|-|+45.7% (vs. Qwen3 VL 8B)|
> |gWorld 32B|79.6|-|+27.1% (vs. Qwen3 VL 32B)|
>
> ### **[Q4] Measuring potential policy gains**
>
> We measured *potential* (including ground truth as a candidate) performance gains to isolate the positive effects of a strong world model.
>
> Concretely, there are two components within the policy system:
> 1. Generating K candidate actions.
> 2. Imagining each next state conditioned on the K candidate actions via the world model or value-function.
>
> The capabilities of the first component are wholly dependent on the agent's `pass@k` rate and therefore *orthogonal* to the effectiveness of the world model itself. If the ground truth is not included by default, the realized policy gain becomes a *moving target*: it will fall between 0% (if the backbone's `pass@k` is 0) and our reported potential values.
>
> ### **[Q5] Quantifying the quality of synthetic reasoning traces**
>
> We conducted a preliminary qualitative analysis of 100 randomly sampled synthetic reasoning traces ($R_t$) and observed a 0% error rate. We achieved this via iterations over different prompts, to elicit faithful and consistent reasoning conditioned on both the current state $S_t$ and the subsequent state $S_{t+1}$.
>
> Beyond this manual verification, our experiments provide strong implicit and explicit validation in **Fig. 5** and **6**.
>
> ### **[Additional Result During Rebuttal]**
>
> Please consider additional results in **[Q5] Our method's end-to-end inference is up to 291x faster than VIMO** in Response to `w2bh`.

---

> > ### Author Rebuttal · Reviewer_D2FN · 2026-04-03
> >
> > Thanks. I keep my score.

---

> > > ### Author Response · Authors · 2026-04-06
> > >
> > > Thank you for your constructive comments and for confirming that our rebuttal has adequately addressed your concerns.
> > >
> > > We will ensure all new results and discussion points are incorporated into the revised manuscript to further improve the paper's clarity.
> > >
> > > Best,
> > > Authors

---

### Decision · Program_Chairs · 2026-04-30

**Decision:**

Accept (regular)

**Comment:**

The paper introduces gWorld, a novel mobile GUI world model that predicts future UI states as renderable web code rather than pixels. This approach elegantly bypasses the text-rendering bottlenecks of diffusion-based models while maintaining high visual fidelity. The authors also introduce an automated data synthesis pipeline and a comprehensive benchmark MWMBENCH.

Following the rebuttal, all reviewers were satisfied. Reviewers maintained or raised their scores to the positive side (Weak Accept/Accept).